# Deep Learning Through A Telescoping Lens: A Simple Model Provides Empirical Insights On Grokking, Gradient Boosting & Beyond

**Alan Jeffares**[*]
University of Cambridge
aj659@cam.ac.uk

**Alicia Curth**[*]
University of Cambridge
amc253@cam.ac.uk

**Mihaela van der Schaar**
University of Cambridge
mv472@cam.ac.uk

## Abstract

Deep learning sometimes *appears to* work in unexpected ways. In pursuit of a deeper understanding of its surprising behaviors, we investigate the utility of a simple yet accurate model of a trained neural network consisting of a sequence of first-order approximations *telescoping* out into a single empirically operational tool for practical analysis. Across three case studies, we illustrate how it can be applied to derive new empirical insights on a diverse range of prominent phenomena in the literature – including double descent, grokking, linear mode connectivity, and the challenges of applying deep learning on tabular data – highlighting that this model allows us to construct and extract metrics that help predict and understand the a priori unexpected performance of neural networks. We also demonstrate that this model presents a pedagogical formalism allowing us to isolate components of the training process even in complex contemporary settings, providing a lens to reason about the effects of design choices such as architecture & optimization strategy, and reveals surprising parallels between neural network learning and gradient boosting.

## 1 Introduction

Deep learning *works*, but it sometimes works in mysterious ways. Despite the remarkable recent success of deep learning in applications ranging from image recognition [KSH12] to text generation [BMR+20], there remain many contexts in which it performs in apparently unpredictable ways: neural networks sometimes exhibit surprisingly non-monotonic generalization performance [BHMM19, PBE+22], continue to be outperformed by gradient boosted trees on tabular tasks despite successes elsewhere [GOV22], and sometimes behave surprisingly similarly to linear models [FDRC20]. The pursuit of a deeper understanding of deep learning and its phenomena has since motivated many subfields, and progress on fundamental questions has been distributed across many distinct yet complementary perspectives that range from purely theoretical to predominantly empirical research.

**Outlook.** In this work, we take a hybrid approach and investigate how we can *apply* ideas primarily used in theoretical research to investigate the behavior of a simple yet accurate model of a neural network *empirically*. Building upon previous work that studies linear approximations to learning in neural networks through tangent kernels (e.g. [JGH18, COB19], see Sec. 2), we consider a model that uses first-order approximations for the functional updates made during training. However, unlike most previous work, we define this model incrementally by simply *telescoping out* approximations to individual updates made during training (Sec. 3) such that it more closely approximates the true behavior of a fully trained neural network in practical settings. This provides us with a pedagogical lens through which we can view modern optimization strategies and other design choices (Sec. 5), and a mechanism with which we can conduct empirical investigations into several prominent deep learning phenomena that showcase how neural networks sometimes generalize *seemingly* unpredictably.

---

[*]Equal contribution

38th Conference on Neural Information Processing Systems (NeurIPS 2024).

Across three case studies in Sec. 4, we then show that this model allows us to *construct and extract metrics that help predict and understand* the a priori unexpected performance of neural networks. First, in Sec. 4.1, we demonstrate that it allows us to extend [CJvdS23]'s recent model complexity metric to neural networks, and use this to investigate surprising generalization curves – discovering that the non-monotonic behaviors observed in both deep double descent [BHMM19] and grokking [PBE+22] are associated with *quantifiable* divergence of train- and test-time model complexity. Second, in Sec. 4.2, we show that it reveals perhaps surprising parallels between gradient boosting [Fri01] and neural network learning, which we then use to investigate the known performance differences between neural networks and gradient boosted trees on tabular data in the presence of dataset irregularities [MKV+23]. Third, in Sec. 4.3, we use it to investigate the connections between gradient stabilization and the success of weight averaging (i.e. linear mode connectivity [FDRC20]).

## 2 Background

**Notation and preliminaries.** Let $f_{\boldsymbol{\theta}} : \mathcal{X} \subseteq \mathbb{R}^d \to \mathcal{Y} \subseteq \mathbb{R}^k$ denote a neural network parameterized by (stacked) model weights $\boldsymbol{\theta} \in \mathbb{R}^p$. Assume we observe a training sample of $n$ input-output pairs $\{\mathbf{x}_i, y_i\}_{i=1}^n$, i.i.d. realizations of the tuple $(X, Y)$ sampled from some distribution $P$, and wish to learn good model parameters $\boldsymbol{\theta}$ for predicting outputs from this data by minimizing an empirical prediction loss $\frac{1}{n} \sum_{i=1}^n \ell(f_{\boldsymbol{\theta}}(\mathbf{x}_i), y_i)$, where $\ell : \mathbb{R}^k \times \mathbb{R}^k \to \mathbb{R}$ denotes some differentiable loss function. Throughout, we let $k = 1$ for ease of exposition, but unless otherwise indicated our discussion generally extends to $k > 1$. We focus on the case where $\boldsymbol{\theta}$ is optimized by initializing the model with some $\boldsymbol{\theta}_0$ and then iteratively updating the parameters through stochastic gradient descent (SGD) with learning rates $\gamma_t$ for $T$ steps, where at each $t \in [T] = \{1, \ldots, T\}$ we subsample batches $B_t \subseteq [n] = \{1, \ldots, n\}$ of the training indices, leading to parameter updates $\Delta \boldsymbol{\theta}_t := \boldsymbol{\theta}_t - \boldsymbol{\theta}_{t-1}$ as:

$$\boldsymbol{\theta}_t = \boldsymbol{\theta}_{t-1} + \Delta \boldsymbol{\theta}_t = \boldsymbol{\theta}_{t-1} - \frac{\gamma_t}{|B_t|} \sum_{i \in B_t} \nabla_{\boldsymbol{\theta}} f_{\boldsymbol{\theta}_{t-1}}(\mathbf{x}_i) g_{it}^{\ell} = \boldsymbol{\theta}_{t-1} - \gamma_t \mathbf{T}_t \mathbf{g}_t^{\ell} \tag{1}$$

where $g_{it}^{\ell} = \frac{\partial \ell(f_{\boldsymbol{\theta}_{t-1}}(\mathbf{x}_i), y_i)}{\partial f_{\boldsymbol{\theta}_{t-1}}(\mathbf{x}_i)}$ is the gradient of the loss w.r.t. the model prediction for the $i^{th}$ training example, which we will sometimes collect in the vector $\mathbf{g}_t^{\ell} = [g_{1t}^{\ell}, \ldots, g_{nt}^{\ell}]^{\top}$, and the $p \times n$ matrix $\mathbf{T}_t = [\frac{\mathbf{1}\{1 \in B_t\}}{|B_t|} \nabla_{\boldsymbol{\theta}} f_{\boldsymbol{\theta}_{t-1}}(\mathbf{x}_1), \ldots, \frac{\mathbf{1}\{n \in B_t\}}{|B_t|} \nabla_{\boldsymbol{\theta}} f_{\boldsymbol{\theta}_{t-1}}(\mathbf{x}_n)]$ has as columns the gradients of the model prediction with respect to its parameters for examples in the training batch (and $\mathbf{0}$ otherwise). Beyond vanilla SGD, modern deep learning practice usually relies on a number of modifications to the update described above, such as momentum and weight decay; we discuss these in Sec. 5.

**Related work: Linearized neural networks and tangent kernels.** A growing body of recent work has explored the use of *linearized* neural networks (linear in their parameters) as a tool for theoretical [JGH18, COB19, LXS+19] and empirical [FDP+20, LZB20, OJMDF21] study. In this paper, we similarly make extensive use of the following observation (as in e.g. [FDP+20]): we can linearize the difference $\Delta f_t(\mathbf{x}) := f_{\boldsymbol{\theta}_t}(\mathbf{x}) - f_{\boldsymbol{\theta}_{t-1}}(\mathbf{x})$ between two parameter updates as

$$\Delta f_t(\mathbf{x}) = \nabla_{\boldsymbol{\theta}} f_{\boldsymbol{\theta}_{t-1}}(\mathbf{x})^{\top} \Delta \boldsymbol{\theta}_t + \mathcal{O}(||\Delta \boldsymbol{\theta}_t||^2) \approx \nabla_{\boldsymbol{\theta}} f_{\boldsymbol{\theta}_{t-1}}(\mathbf{x})^{\top} \Delta \boldsymbol{\theta}_t := \Delta \tilde{f}_t(\mathbf{x}) \tag{2}$$

where the quality of the approximation $\Delta \tilde{f}_t(\mathbf{x})$ is good whenever the parameter updates $\Delta \boldsymbol{\theta}_t$ from a single batch are sufficiently small (or when the Hessian product $||\Delta \boldsymbol{\theta}_t^{\top} \nabla_{\boldsymbol{\theta}}^2 f_{\boldsymbol{\theta}_{t-1}}(\mathbf{x}) \Delta \boldsymbol{\theta}_t||$ vanishes). If Eq. (2) holds exactly (e.g. for infinitesimal $\gamma_t$), then running SGD in the network's parameter space to obtain $\Delta \boldsymbol{\theta}_t$ corresponds to executing steepest descent on the function output $f_{\boldsymbol{\theta}}(\mathbf{x})$ itself using the *neural tangent kernel* $K_t^{\boldsymbol{\theta}}(\mathbf{x}, \mathbf{x}_i)$ at time-step $t$ [JGH18], i.e. results in functional updates

$$\Delta \tilde{f}_t(\mathbf{x}) \approx -\gamma_t \sum_{i \in [n]} K_t^{\boldsymbol{\theta}}(\mathbf{x}, \mathbf{x}_i) g_{it}^{\ell} \text{ where } K_t^{\boldsymbol{\theta}}(\mathbf{x}, \mathbf{x}_i) := \frac{\mathbf{1}\{i \in B_t\}}{|B_t|} \nabla_{\boldsymbol{\theta}} f_{\boldsymbol{\theta}_{t-1}}(\mathbf{x})^{\top} \nabla_{\boldsymbol{\theta}} f_{\boldsymbol{\theta}_{t-1}}(\mathbf{x}_i). \tag{3}$$

*Lazy learning* [COB19] occurs as the model gradients remain approximately constant during training, i.e. $\nabla_{\boldsymbol{\theta}} f_{\boldsymbol{\theta}_t}(\mathbf{x}) \approx \nabla_{\boldsymbol{\theta}} f_{\boldsymbol{\theta}_0}(\mathbf{x})$, $\forall t \in [T]$. For learned parameters $\boldsymbol{\theta}_T$, this implies that the approximation $f_{\boldsymbol{\theta}_T}^{lin}(\mathbf{x}) = f_{\boldsymbol{\theta}_0}(\mathbf{x}) + \nabla_{\boldsymbol{\theta}} f_{\boldsymbol{\theta}_0}(\mathbf{x})^{\top} (\boldsymbol{\theta}_T - \boldsymbol{\theta}_0)$ holds – which is a *linear function of the model parameters*, and thus corresponds to a linear regression in which features are given by the model gradients $\nabla_{\boldsymbol{\theta}} f_{\boldsymbol{\theta}_0}(\mathbf{x})$ instead of the inputs $\mathbf{x}$ directly – whose training dynamics can be more easily understood theoretically. For sufficiently wide neural networks the $\nabla_{\boldsymbol{\theta}} f_{\boldsymbol{\theta}_t}(\mathbf{x})$, and thus the tangent kernel, have been theoretically shown to be constant throughout training in some settings [JGH18, LXS+19], but in practice they generally vary during training, as shown theoretically in [LZB20] and empirically in [FDP+20]. A growing theoretical literature [GPK22] investigates constant tangent kernel assumptions to study convergence and generalization of neural networks (e.g.

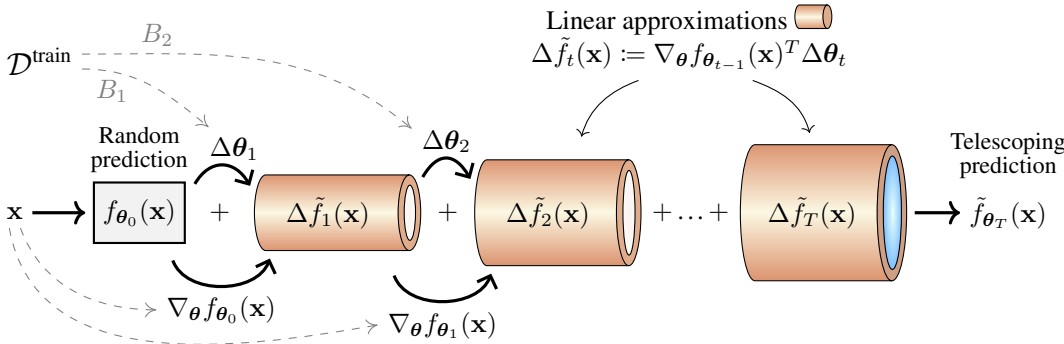

Figure 1: **Illustration of the telescoping model of a trained neural network.** Unlike the more standard framing of a neural network in terms of an iteratively learned set of parameters, the telescoping model takes a functional perspective on training a neural network in which an arbitrary test example's initially random prediction, $f_{\boldsymbol{\theta}_0}(\mathbf{x})$, is additively updated by a linearized adjustment $\Delta \tilde{f}_t(\mathbf{x})$ at each step $t$ as in Eq. (5).

[JGH18, LXS+19, DLL+19, BM19, GMMM19, GSJW20]). This present work relates more closely to *empirical* studies making use of tangent kernels and linear approximations, such as [LSP+20, OJMDF21] who highlight differences between lazy learning and real networks, and [FDP+20] who empirically investigate the relationship between loss landscapes and the evolution of $K_t^{\boldsymbol{\theta}}(\mathbf{x}, \mathbf{x}_i)$.

## 3   A Telescoping Model of Deep Learning

In this work, we explore whether we can exploit the approximation in Eq. (2) beyond the laziness assumption to gain new insight into neural network learning. Instead of applying the approximation across the entire training trajectory at once as in $f_{\boldsymbol{\theta}_T}^{lin}(\mathbf{x})$, we consider using it *incrementally* at each batch update during training to approximate *what has been learned at this step*. This still provides us with a greatly simplified and transparent model of a neural network, and results in a much more reasonable approximation of the true network. Specifically, we explore whether – instead of studying the final model $f_{\boldsymbol{\theta}_T}(\mathbf{x})$ as a whole – we can gain insight by *telescoping out* the functional updates made throughout training, i.e. exploiting that we can always equivalently express $f_{\boldsymbol{\theta}_T}(\mathbf{x})$ as:

$$f_{\boldsymbol{\theta}_T}(\mathbf{x}) = f_{\boldsymbol{\theta}_0}(\mathbf{x}) + \sum_{t=1}^{T}[f_{\boldsymbol{\theta}_t}(\mathbf{x}) - f_{\boldsymbol{\theta}_{t-1}}(\mathbf{x})] = f_{\boldsymbol{\theta}_0}(\mathbf{x}) + \sum_{t=1}^{T} \Delta f_t(\mathbf{x}) \tag{4}$$

This representation of a trained neural network in terms of its learning trajectory rather than its final parameters is interesting because we are able to better reason about the impact of the training procedure on the intermediate updates $\Delta f_t(\mathbf{x})$ than the final function $f_{\boldsymbol{\theta}_T}(\mathbf{x})$ itself. In particular, we investigate whether empirically monitoring behaviors of the sum in Eq. (4) while making use of the approximation in Eq. (2) will enable us to gain practical insights into learning in neural networks, while incorporating a variety of modern design choices into the training process. That is, we explore the use of the following *telescoping model* $\tilde{f}_{\boldsymbol{\theta}_T}(\mathbf{x})$ as an approximation of a trained neural network:

---
**Telescoping model of a trained neural network**

$$\tilde{f}_{\boldsymbol{\theta}_T}(\mathbf{x}) := f_{\boldsymbol{\theta}_0}(\mathbf{x}) + \sum_{t=1}^{T} \underbrace{\nabla_{\boldsymbol{\theta}} f_{\boldsymbol{\theta}_{t-1}}(\mathbf{x})^{\top} \Delta \boldsymbol{\theta}_t}_{\substack{\text{(i) The \textit{weight-averaging}} \\ \text{representation}}} = f_{\boldsymbol{\theta}_0}(\mathbf{x}) - \sum_{t=1}^{T} \underbrace{\sum_{i \in [n]} K_t^T(\mathbf{x}, \mathbf{x}_i) g_{it}^{\ell}}_{\text{(ii) The \textit{kernel} representation}} \tag{5}$$

---

where $K_t^T(\mathbf{x}, \mathbf{x}_i)$ is determined by the neural tangent kernel as $\gamma_t K_t^{\boldsymbol{\theta}}(\mathbf{x}, \mathbf{x}_i)$ in the case of standard SGD (in which case (ii) can also be interpreted as a discrete-time approximation of [Dom20]'s *path kernel*), but can take other forms for different choices of learning algorithm as we explore in Sec. 5.

**Practical considerations.** Before proceeding, it is important to emphasize that the telescoping approximation described in Eq. (5) is intended as *a tool for (empirical) analysis of learning in neural networks* and is *not* being proposed as an alternative approach to training neural networks. Obtaining $\tilde{f}_{\boldsymbol{\theta}_T}(\mathbf{x})$ requires computing $\nabla_{\boldsymbol{\theta}} f_{\boldsymbol{\theta}_{t-1}}(\mathbf{x})$ for each training *and* testing example at each training step $t \in [T]$, leading to increased computation over standard training. Additionally, these computational costs are likely prohibitive for extremely large networks and datasets without further adjustments; for this purpose, further approximations such as [MBS23] could be explored. Nonetheless, computing $\tilde{f}_{\boldsymbol{\theta}_T}(\mathbf{x})$ – or relevant parts of it – is still feasible in many pertinent settings as later illustrated in Sec. 4.

**How good is this approximation?** In Fig. 2, we examine the quality of $\tilde{f}_{\boldsymbol{\theta}_t}(\mathbf{x})$ for a 3-layer fully-connected ReLU network of width 200, trained to discriminate 3-vs-5 from 1000 MNIST examples using the squared loss with SGD or AdamW [LH17]. In red, we plot its mean average approximation error ($\frac{1}{1000}\sum_{\mathbf{x}\in\mathcal{X}_{test}}|f_{\boldsymbol{\theta}_t}(\mathbf{x}) - \tilde{f}_{\boldsymbol{\theta}_t}(\mathbf{x})|$) and observe that for small learning rates $\gamma$ the difference remains negligible. In gray we plot the same quantity for $f_{\boldsymbol{\theta}_t}^{lin}(\mathbf{x})$ (i.e. the first-order expansion around $\boldsymbol{\theta}_0$) for reference and find that iteratively telescoping out the updates instead improves the approximation *by orders of magnitude* – which is also reflected in their prediction performance (see Appendix D.1). Unsurprisingly, $\gamma$ controls approximation quality as it determines $||\Delta\boldsymbol{\theta}_t||$. Further, $\gamma$ *interacts* with the optimizer choice – e.g. Adam(W) [KB14, LH17] naturally makes larger updates due to rescaling (see Sec. 5) and therefore requires smaller $\gamma$ to ensure approximation quality than SGD.

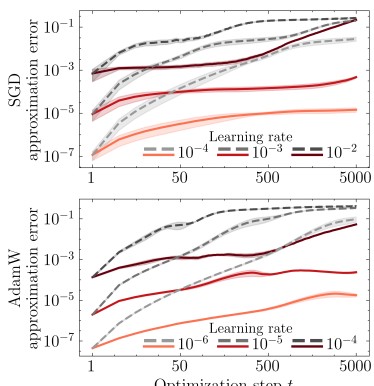

Figure 2: **Approximation error** of the telescoping ($\tilde{f}_{\boldsymbol{\theta}_t}(\mathbf{x})$, red) and the linear model ($f_{\boldsymbol{\theta}_t}^{lin}(\mathbf{x})$, gray).

## 4 A Closer Look at Deep Learning Phenomena Through a Telescoping Lens

Next, we turn to *applying* the telescoping model. Below, we present three case studies revisiting existing experiments that provided evidence for a range of unexpected behaviors of neural networks. These case studies have in common that they highlight cases in which neural networks appear to generalize somewhat *unpredictably*, which is also why each phenomenon has received considerable attention in recent years. For each, we then show that the telescoping model allows us to construct and extract metrics that can help predict and understand the unexpected performance of the networks. In particular, we investigate (i) surprising generalization curves (Sec. 4.1), (ii) performance differences between gradient boosting and neural networks on some tabular tasks (Sec. 4.2), and (iii) the success of weight averaging (Sec. 4.3). We include an extended literature review in Appendix A, a detailed discussion of all experimental setups in Appendix C, and additional results in Appendix D.

### 4.1 Case study 1: Exploring surprising generalization curves and benign overfitting

Classical statistical wisdom provides clear intuitions about overfitting: models that can fit the training data too well – because they have too many parameters and/or because they were trained for too long – are expected to generalize poorly (e.g. [HTF09, Ch. 7]). Modern phenomena like double descent [BHMM19], however, highlighted that pure capacity measures (capturing what *could* be learned instead of what *is* actually learned) would not be sufficient to understand the complexity-generalization relationship in deep learning [Bel21]. Raw parameter counts, for example, cannot be enough to understand the complexity of what has been learned by a neural network during training because, even when using *the same architecture*, what is learned could be wildly different across various implementation choices within the optimization process – and even at different points during the training process of the same model, as prominently exemplified by the grokking phenomenon [PBE+22]. Here, with the goal of finding clues that may help predict phenomena like double descent and grokking, we explore whether the telescoping model allows us to gain insight into the relative complexity of what is learned.

A complexity measure that avoids the shortcomings listed above – because it allows to consider a *specific trained* model – was recently used by [CJvdS23] in their study of *non-deep* double descent. As their measure $p_{\hat{\mathbf{s}}}^0$ builds on the literature on smoothers [HT90], it requires to express learned predictions as a linear combination of the training labels, i.e. as $f(\mathbf{x}) = \hat{\mathbf{s}}(\mathbf{x})\mathbf{y} = \sum_{i\in[n]}\hat{s}^i(\mathbf{x})y_i$. Then, [CJvdS23] define the *effective parameters* $p_{\hat{\mathbf{s}}}^0$ used by the model when issuing predictions for some set of inputs $\{\mathbf{x}_j^0\}_{j\in\mathcal{I}_0}$ with indices collected in $\mathcal{I}_0$ (here, $\mathcal{I}_0$ is either $\mathcal{I}_{train} = \{1,\ldots,n\}$ or $\mathcal{I}_{test} = \{n+1,\ldots,n+m\}$) as $p_{\hat{\mathbf{s}}}^0 \equiv p(\mathcal{I}_0, \hat{\mathbf{s}}(\cdot)) = \frac{n}{|\mathcal{I}_0|}\sum_{j\in\mathcal{I}_0}||\hat{\mathbf{s}}(\mathbf{x}_j^0)||^2$. Intuitively, the larger $p_{\hat{\mathbf{s}}}^0$, the less smoothing across the training labels is performed, which implies higher model complexity.

Due to the black-box nature of trained neural networks, however, it is not obvious how to link learned predictions to the labels observed during training. Here, we demonstrate how the telescoping model allows us to do precisely that – enabling us to make use of $p_{\hat{\mathbf{s}}}^0$ as a proxy for complexity. We consider the special case of a single output ($k=1$) and training with squared loss $\ell(f(\mathbf{x}), y) = \frac{1}{2}(y - f(\mathbf{x}))^2$,

and note that we can now exploit that the SGD weight update simplifies to

$$\Delta\boldsymbol{\theta}_t = \gamma_t \mathbf{T}_t(\mathbf{y} - \mathbf{f}_{\boldsymbol{\theta}_{t-1}}) \text{ where } \mathbf{y} = [y_1, \dots, y_n]^\top \text{ and } \mathbf{f}_{\boldsymbol{\theta}_t} = [f_{\boldsymbol{\theta}_t}(\mathbf{x}_1), \dots, f_{\boldsymbol{\theta}_t}(\mathbf{x}_n)]^\top. \quad (6)$$

Assuming the telescoping approximation holds exactly, this implies functional updates

$$\Delta\tilde{f}_t(\mathbf{x}) = \gamma_t \nabla_{\boldsymbol{\theta}} f_{\boldsymbol{\theta}_{t-1}}(\mathbf{x})^\top \mathbf{T}_t(\mathbf{y} - \tilde{\mathbf{f}}_{\boldsymbol{\theta}_{t-1}}) \quad (7)$$

which use a linear combination of the training labels. Note further that after the *first* SGD update

$$\tilde{f}_{\boldsymbol{\theta}_1}(\mathbf{x}) = f_{\boldsymbol{\theta}_0}(\mathbf{x}) + \Delta\tilde{f}_1(\mathbf{x}) = \underbrace{\gamma_1 \nabla_{\boldsymbol{\theta}} f_{\boldsymbol{\theta}_0}(\mathbf{x})^\top \mathbf{T}_1}_{\mathbf{s}_{\boldsymbol{\theta}_1}(\mathbf{x})} \mathbf{y} + \underbrace{f_{\boldsymbol{\theta}_0}(\mathbf{x}) - \gamma_1 \nabla_{\boldsymbol{\theta}} f_{\boldsymbol{\theta}_0}(\mathbf{x})^\top \mathbf{T}_1 \mathbf{f}_{\boldsymbol{\theta}_0}}_{c_{\boldsymbol{\theta}_1}^0(\mathbf{x})} \quad (8)$$

which means that the first telescoping predictions $\tilde{f}_{\boldsymbol{\theta}_1}(\mathbf{x})$ are indeed simply linear combinations of the training labels (and the predictions at initialization)! As detailed in Appendix B.1, this also implies that recursively substituting Eq. (7) into Eq. (5) further allows us to write *any* prediction $\tilde{f}_{\boldsymbol{\theta}_t}(\mathbf{x})$ as a linear combination of the training labels and $f_{\boldsymbol{\theta}_0}(\cdot)$, i.e. $\tilde{f}_{\boldsymbol{\theta}_t}(\mathbf{x}) = \mathbf{s}_{\boldsymbol{\theta}_t}(\mathbf{x})\mathbf{y} + c_{\boldsymbol{\theta}_t}^0(\mathbf{x})$ where the $1 \times n$ vector $\mathbf{s}_{\boldsymbol{\theta}_t}(\mathbf{x})$ is a function of the kernels $\{K_{t'}^t(\cdot, \cdot)\}_{t' \leq t}$, and the scalar $c_{\boldsymbol{\theta}_t}^0(\mathbf{x})$ is a function of the $\{K_{t'}^t(\cdot, \cdot)\}_{t' \leq t}$ and $f_{\boldsymbol{\theta}_0}(\cdot)$. We derive precise expressions for $\mathbf{s}_{\boldsymbol{\theta}_t}(\mathbf{x})$ and $c_{\boldsymbol{\theta}_t}^0(\mathbf{x})$ for different optimizers in Appendix B.1 – enabling us to use $\mathbf{s}_{\boldsymbol{\theta}_t}(\mathbf{x})$ to compute $p_{\hat{\mathbf{s}}}^0$ as a proxy for complexity below.

**Double descent: Model complexity vs model size.** While training error always monotonically decreases as model size (measured by parameter count) increases, [BHMM19] made a surprising observation regarding test error in their seminal paper on *double descent*: they found that test error initially improves with additional parameters and then worsens when the model is increasingly able to overfit to the training data (as is expected) but can *improve again* as model size is increased further past the so-called interpolation threshold where perfect training performance is achieved. This would appear to contradict the classical U-shaped relationship between model complexity and test error [HTF09, Ch. 7]. Here, we investigate whether tracking $p_{\hat{\mathbf{s}}}^0$ on train and test data separately will allow us to gain new insight into the phenomenon in neural networks.

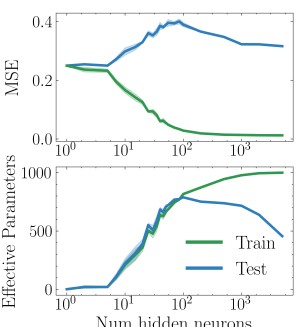

Figure 3: **Double descent** in MSE (top) and effective parameters $p_{\hat{\mathbf{s}}}^0$ (bottom) on CIFAR-10.

In Fig. 3, we replicate the binary classification example of double descent in neural networks of [BHMM19], training single-hidden-layer ReLU networks of increasing width to distinguish cats and dogs on CIFAR-10 (we present additional results using MNIST in Appendix D.2). First, we indeed observe the characteristic behavior of error curves as described in [BHMM19] (top panel). Measuring learned complexity using $p_{\hat{\mathbf{s}}}^0$, we then find that while $p_{\hat{\mathbf{s}}}^{train}$ monotonically increases as model size is increasing, the effective parameters used on the test data $p_{\hat{\mathbf{s}}}^{test}$ implied by the trained neural network *decrease* as model size is increased past the interpolation threshold (bottom panel). Thus, paralleling the findings made in [CJvdS23] for linear regression and tree-based methods, we find that distinguishing between train- and test-time complexity of a neural network using $p_{\hat{\mathbf{s}}}^0$ provides new quantitative evidence that bigger networks are *not* necessarily learning more complex prediction functions for unseen test examples, which resolves the ostensible tension between deep double descent and the classical U-curve. Importantly, note that $p_{\hat{\mathbf{s}}}^{test}$ can be computed without access to test-time labels, which means that the observed difference between $p_{\hat{\mathbf{s}}}^{train}$ and $p_{\hat{\mathbf{s}}}^{test}$ allows to quantify whether there is *benign overfitting* [BLLT20, YHT+21] in a neural network.

**Grokking: Model complexity throughout training.** The grokking phenomenon [PBE+22] then showcased that improvements in test performance *during a single training run* can occur long after perfect training performance has been achieved (contradicting early stopping practice!). While [LMT22] attribute this to weight decay causing $||\boldsymbol{\theta}_t||$ to shrink late in training – which they demonstrate on an MNIST example using unusually large $\boldsymbol{\theta}_0$ – [KBGP24] highlight that grokking can also occur as the weight norm $||\boldsymbol{\theta}_t||$ *grows* later in training – which they demonstrate on a polynomial regression task. In Fig. 4 we replicate[2] both experiments while tracking $p_{\hat{\mathbf{s}}}^0$ to investigate whether

---

[2]As detailed in Appendix C, we replicate [KBGP24]'s experiment exactly but adapt [LMT22]'s experiment into a binary classification task with lower learning rate $\gamma$ to enable the use of $\tilde{f}_{\boldsymbol{\theta}_T}(\mathbf{x})$. The reduction of $\gamma$ is needed here as the $\Delta\boldsymbol{\theta}_t$ are otherwise too large to obtain an accurate approximation and has a side effect that the grokking phenomenon appears visually less extreme as perfect training performance is achieved later in training.

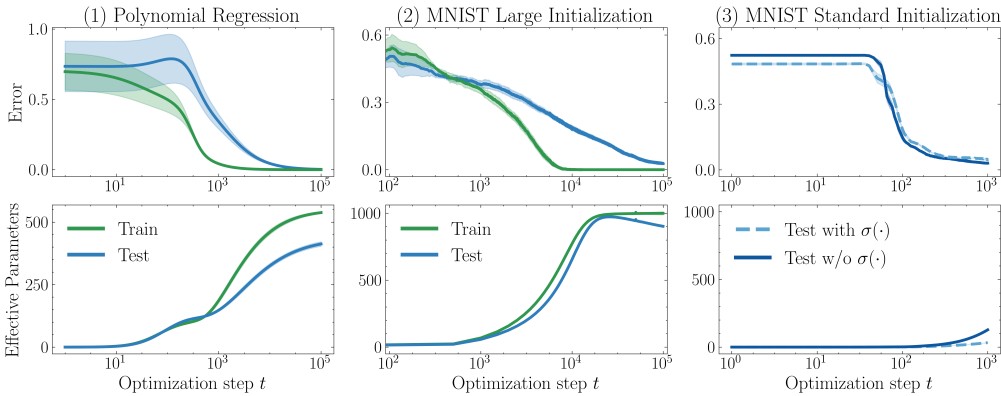

Figure 4: **Grokking** in mean squared error on a polynomial regression task (1, replicated from [KBGP24]) and in misclassification error on MNIST using a network with large initialization (2, replicated from [LMT22]) (top), against effective parameters (bottom). Column (3) shows test results on MNIST with standard initialization (with and without sigmoid activation) where time to generalization is quick and grokking does not occur.

this provides new insight into this apparent disagreement. Then, we observe that the continued improvement in test error, past the point of perfect training performance, is associated with divergence of $p_{\hat{\mathbf{s}}}^{train}$ and $p_{\hat{\mathbf{s}}}^{test}$ in *both* experiments (analogous to the double descent experiment in Fig. 3), suggesting that grokking may reflect *transition into* a measurably benign overfitting regime during training. In Appendix D.2, we additionally investigate mechanisms known to induce grokking, and show that later onset of generalization indeed coincides with later divergence of $p_{\hat{\mathbf{s}}}^{train}$ and $p_{\hat{\mathbf{s}}}^{test}$.

**Inductive biases & learned complexity.** We observed that the large $\boldsymbol{\theta}_0$ in [LMT22]'s MNIST example of grokking result in very large initial predictions $|f_{\boldsymbol{\theta}_0}(\mathbf{x})| \gg 1$. Because *no* sigmoid is applied, the model needs to learn that all $y_i \in [0, 1]$ by reducing the magnitude of predictions substantially – large $\boldsymbol{\theta}_0$ thus constitute a very poor inductive bias for this task. One may expect that the better an inductive bias is, the less complex the component of the final prediction that is learned from data. To test whether this intuition is quantifiable, we repeat the MNIST experiment with standard initialization scale, with and without sigmoid activation $\sigma(\cdot)$, in column (3) of Fig. 4 (training results shown in Appendix D.2 for readability). We indeed find that both not only speed up learning significantly (a generalizing solution is found in $10^2$ instead of $10^5$ steps), but also substantially reduce effective parameters used, where the stronger inductive bias – using $\sigma(\cdot)$ – indeed leads to the least learned complexity.

***Takeaway Case Study 1.*** The telescoping model enables us to use $p_{\hat{\mathbf{s}}}^0$ as a proxy for learned complexity, whose relative behavior on train and test data can quantify benign overfitting in neural networks.

## 4.2 Case study 2: Understanding differences between gradient boosting and neural networks

Despite their overwhelming successes on image and language data, neural networks are – perhaps surprisingly – still widely considered to be outperformed by *gradient boosted trees* (GBTs) on *tabular data*, an important modality in many data science applications. Exploring this apparent Achilles heel of neural networks has therefore been the goal of multiple extensive benchmarking studies [GOV22, MKV+23]. Here, we concentrate on a specific empirical finding of [MKV+23]: their results suggest that GBTs may particularly outperform deep learning on heterogeneous data with greater irregularity in input features, a characteristic often present in tabular data. Below, we first show that the telescoping model offers a useful lens to compare and contrast the two methods, and then use this insight to provide and test a new explanation of why GBTs can perform better in the presence of dataset irregularities.

**Identifying (dis)similarities between learning in GBTs and neural networks.** We begin by introducing gradient boosting [Fri01] closely following [HTF09, Ch. 10.10]. Gradient boosting (GB) also aims to learn a predictor $\hat{f}^{GB} : \mathcal{X} \rightarrow \mathbb{R}^k$ minimizing expected prediction loss $\ell$. While deep learning solves this problem by iteratively updating a randomly initialized set of *parameters* that transform inputs to predictions, the GB formulation iteratively updates *predictions* directly without requiring any iterative learning of parameters – thus operating in function space rather than parameter space. Specifically, GB, with learning rate $\gamma$ and initialized at predictor $h_0(\mathbf{x})$, consists of a sequence $\hat{f}_T^{GB}(\mathbf{x}) = h_0(\mathbf{x}) + \gamma \sum_{t=1}^{T} \hat{h}_t(\mathbf{x})$ where each $\hat{h}_t(\mathbf{x})$ improves upon the existing predictions $\hat{f}_{t-1}^{GB}(\mathbf{x})$. The solution to the loss minimization problem can be achieved by executing steepest descent in function space *directly*, where each update $\hat{h}_t$ simply outputs the negative training gradients of the loss function with respect to the previous model, i.e. $\hat{h}_t(\mathbf{x}_i) = -g_{it}^{\ell}$ where $g_{it}^{\ell} = \partial \ell(\hat{f}_{t-1}^{GB}(\mathbf{x}_i), y_i)/\partial \hat{f}_{t-1}^{GB}(\mathbf{x}_i)$.

However, this process is only defined at the training points $\{\mathbf{x}_i, y_i\}_{i\in[n]}$. To obtain an estimate of the loss gradient for an arbitrary test point $\mathbf{x}$, each iterative update instead fits a weak learner $\hat{h}_t(\cdot)$ to the current input-gradient pairs $\{\mathbf{x}_i, -g_{it}^\ell\}_{i\in[n]}$ which can then also be evaluated new, unseen inputs. While this process could in principle be implemented using any base learner, the term *gradient boosting* today appears to exclusively refer to the approach outlined above implemented using shallow trees as $\hat{h}_t(\cdot)$ [Fri01]. Focusing on trees which issue predictions by averaging the training outputs in each leaf, we can make use of the fact that these are sometimes interpreted as adaptive nearest neighbor estimators or kernel smoothers [LJ06, BD10, CJvdS24], allowing us to express the learned predictor as:

$$\hat{f}^{GB}(\mathbf{x}) = h_0(\mathbf{x}) - \gamma \sum_{t=1}^{T} \sum_{i\in[n]} \frac{\mathbf{1}\{l_{h_t}(\mathbf{x}) = l_{h_t}(\mathbf{x}_i)\}}{n_{l(\mathbf{x})}} g_{it}^\ell = h_0(\mathbf{x}) - \gamma \sum_{t=1}^{T} \sum_{i\in[n]} K_{\hat{h}_t}(\mathbf{x}, \mathbf{x}_i) g_{it}^\ell \quad (9)$$

where $l_{\hat{h}_t}(\mathbf{x})$ denotes the leaf example $\mathbf{x}$ falls into, $n_{l(\mathbf{x})} = \sum_{i\in[n]} \mathbf{1}\{l_{h_t}(\mathbf{x}) = l_{h_t}(\mathbf{x}_i)\}$ is the number of training examples in said leaf and $K_{\hat{h}_t}(\mathbf{x}, \mathbf{x}_i) = {}^1/n_{leaf(\mathbf{x})}\mathbf{1}\{l_{\hat{h}_t}(\mathbf{x}) = l_{\hat{h}_t}(\mathbf{x}_i)\}$ is thus the kernel learned by the $t^{th}$ tree $\hat{h}_t(\cdot)$. Comparing Eq. (9) to the kernel representation of the telescoping model of neural network learning in Eq. (5), we make a perhaps surprising observation: the telescoping model of a neural network and GBTs have *identical* structure and differ only in their used kernel! Below, we explore whether this new insight allows to understand some of their performance differences.

**Why can GBTs outperform deep learning in the presence of dataset irregularities?** Comparing Eq. (5) and Eq. (9) thus suggests that at least some of the performance differences between neural networks and GBTs are likely to be rooted in the differences between the behavior of the neural network tangent kernels $K_t^{\boldsymbol{\theta}}(\mathbf{x}, \mathbf{x}_i)$ and GBT's tree kernels $K_{\hat{h}_t}(\mathbf{x}, \mathbf{x}_i)$. One difference is obvious and purely architectural: it is possible that either kernel encodes a better inductive bias to fit the underlying outcome-generating process of a dataset at hand. Another difference is more subtle and relates to the behavior of the learned model on new inputs $\mathbf{x}$: the tree kernels are likely to behave *much* more predictable at test-time than the neural network tangent kernels. To see this, note that for the tree kernels we have that $\forall \mathbf{x} \in \mathcal{X}$ and $\forall i \in [n]$, $0 \leq K_{\hat{h}_t}(\mathbf{x}, \mathbf{x}_i) \leq 1$ and $\sum_{i\in[n]} K_{\hat{h}_t}(\mathbf{x}, \mathbf{x}_i) = 1$; importantly, this is true regardless of whether $\mathbf{x} = \mathbf{x}_i$ for some $i$ or not. For the tangent kernels on the other hand, $K_t^{\boldsymbol{\theta}}(\mathbf{x}, \mathbf{x}_i)$ is in general unbounded and could behave *very* differently for $\mathbf{x}$ not observed during training. This leads us to hypothesize that this difference may be able to explain [MKV$^+$23]'s observation that GBTs perform better whenever features are heavy-tailed: if a test point $\mathbf{x}$ is very different from training points, the kernels implied by the neural network $\mathbf{k}_t^{\boldsymbol{\theta}}(\mathbf{x}) \coloneqq [K_t^{\boldsymbol{\theta}}(\mathbf{x}, \mathbf{x}_1), \ldots, K_t^{\boldsymbol{\theta}}(\mathbf{x}, \mathbf{x}_n)]^\top$ may behave very differently than at train-time while the tree kernels $\mathbf{k}_{\hat{h}_t}(\mathbf{x}) \coloneqq [K_{\hat{h}_t}(\mathbf{x}, \mathbf{x}_1), \ldots, K_{\hat{h}_t}(\mathbf{x}, \mathbf{x}_n)]^\top$ will be less affected. For instance, $\frac{1}{\sqrt{n}} \leq ||\mathbf{k}_{\hat{h}_t}(\mathbf{x})||_2 \leq 1$ for all $\mathbf{x}$ while $||\mathbf{k}_t^{\boldsymbol{\theta}}(\mathbf{x})||_2$ is generally unbounded.

We empirically test this hypothesis on standard tabular benchmark datasets proposed in [GOV22]. We wish to examine the performance of the models and the behavior of the kernels as inputs become increasingly irregular, evaluating if GBT's kernels indeed display more consistent behavior compared to the network's tangent kernels. As a simple notion for input irregularity, we apply principal component analysis to the inputs to obtain a lower dimensional representation of the data and sort the observations according to their distance from the centroid. For a fixed trained model, we then evaluate on test sets consisting of increasing proportions $p$ of the most irregular inputs (those in the top 10% furthest from the centroid). We compare the GBTs to neural networks by examining (i) the most extreme values their kernel weights take at test-time relative to the training data (measured as $\frac{\frac{1}{T}\sum_{t=1}^{T} \max_{j\in\mathcal{I}_{test}^p} ||\mathbf{k}_t(x_j)||_2}{\frac{1}{T}\sum_{t=1}^{T} \max_{i\in\mathcal{I}_{train}} ||\mathbf{k}_t(\mathbf{x}_i)||_2}$) and (ii) how their relative mean squared error (measured as $\frac{MSE_{NN}^p - MSE_{GBT}^p}{MSE_{NN}^0 - MSE_{GBT}^0}$) changes as the proportion $p$ of irregular examples increases. In Fig. 5 using `houses` and in Appendix D.3

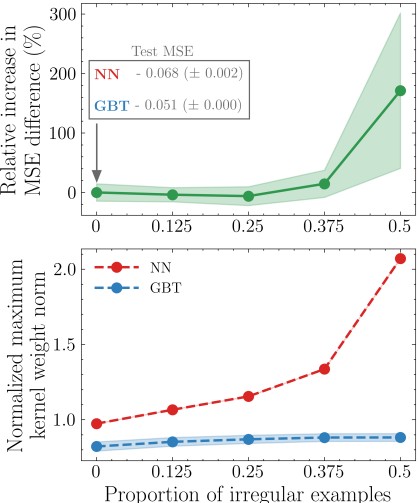

Figure 5: **Neural Networks vs GBTs:** Relative performance (top) and behavior of kernels (bottom) with increasing test data irregularity using the `houses` dataset.

using additional datasets, we first observe that GBTs outperform the neural network already in the absence of irregular examples; this highlights that there may indeed be differences in the suitability of the kernels in fitting the outcome-generating processes. Consistent with our expectations, we then find that, as the test data becomes more irregular, the performance of the neural network decays faster than that of the GBTs. Importantly, this is well tracked by their kernels, where the unbounded nature of the network's tangent kernel indeed results in it changing its behavior on new, challenging examples.

***Takeaway Case Study 2.*** Eq. (5) provides a new lens for comparing neural networks to GBTs, and highlights that unboundedness in $\mathbf{k}_t^{\theta}(\mathbf{x})$ can predict performance differences due to dataset irregularities.

### 4.3 Case study 3: Towards understanding the success of weight averaging

The final interesting phenomenon we investigate is that it is sometimes possible to simply average the weights $\theta_1$ and $\theta_2$ obtained from two stochastic training runs of the same model, resulting in a weight-averaged model that performs no worse than the individual models [FDRC20, AHS22] – which has important applications in areas such as federated learning. This phenomenon is known as linear mode connectivity (LMC) and is surprising as, a priori, it is not obvious that simply *averaging the weights* of independent neural networks (instead of their predictions, as in a deep ensemble [LPB17]), which are highly nonlinear functions of their parameters, would not greatly worsen performance. While recent work has demonstrated empirically that it is sometimes possible to weight-average an even broader class of models after permuting weights [SJ20, ESSN21, AHS22], we focus here on understanding when LMC can be achieved for two models trained from the same initialization $\theta_0$.

In particular, we are interested in [FDRC20]'s observation that LMC can emerge during training: the weights of two models $\theta_{jT}^{t'}, j \in \{1, 2\}$, which are initialized identically and follow identical optimization routine up until checkpoint $t'$ but receive different batch orderings and data augmentations after $t'$, can be averaged to give an equally performant model as long as $t'$ exceeds a so-called *stability point* $t^*$, which was empirically discovered to occur early in training in [FDRC20]. Interestingly, [FDP$^+$20, Sec. 5] implicitly hint at an explanation for this phenomenon in their empirical study of tangent kernels and loss landscapes, where they found an association between the disappearance of loss barriers between solutions during training and the rate of change in $K_t^{\theta}(\cdot, \cdot)$. We further explore potential implications of this observation through the lens of the telescoping model below.

**Why a transition into a constant-gradient regime would imply LMC.** Using the weight-averaging representation of the telescoping model, it becomes easy to see that not only would stabilization of the tangent kernel be *associated* with lower linear loss barriers, but the transition into a lazy regime during training – i.e. reaching a point $t^*$ after which the model gradients no longer change – can be *sufficient* to imply LMC during training as observed in [FDRC20] under a mild assumption on the performance of the two networks' *ensemble*. To see this, let $L(f) := \mathbb{E}_{X,Y \sim P}[\ell(f(X), Y)]$ denote the expected loss of $f$ and recall that if $sup_{\alpha \in [0,1]} L(f_{\alpha\theta_{1T}^{t'} + (1-\alpha)\theta_{2T}^{t'}}) - [\alpha L(f_{\theta_{1T}^{t'}}) + (1 - \alpha)L(f_{\theta_{2T}^{t'}})] \leq 0$ then LMC is said to hold. If we assume that ensembles $\bar{f}^{\alpha}(\mathbf{x}) := \alpha f_{\theta_{1T}^{t'}}(\mathbf{x}) + (1-\alpha)f_{\theta_{2T}^{t'}}(\mathbf{x})$ perform no worse than the individual models (i.e. $L(\bar{f}^{\alpha}) \leq \alpha L(f_{\theta_{1T}^{t'}}) + (1-\alpha)L(f_{\theta_{2T}^{t'}}) \forall \alpha \in [0, 1]$, as is usually the case in practice [ABPC23]), then one case in which LMC is guaranteed is if the predictions of weight-averaged model and ensemble are identical. In Appendix B.2, we show that if there exists some $t^* \in [0, T)$ after which the model gradients $\nabla_{\theta} f_{\theta_{jt}^{t*}}(\cdot)$ no longer change (i.e. for all $t' \geq t^*$ the learned updates $\theta_{jt}^{t'}$ lie in a convex set $\Theta_j^{stable}$ in which $\nabla_{\theta} f_{\theta_{jt}^{t'}}(\cdot) \approx \nabla_{\theta} f_{\theta_{t^*}}(\cdot)$), then indeed

$$\bar{f}^{\alpha}(\mathbf{x}) \approx f_{\alpha\theta_{1T}^{t'} + (1-\alpha)\theta_{2T}^{t'}}(\mathbf{x}) \approx f_{\theta_{t'}}(\mathbf{x}) + \nabla_{\theta} f_{\theta_{t^*}}(\mathbf{x})^{\top} \sum_{t=t'+1}^{T} (\alpha\Delta\theta_{1t}^{t'} + (1 - \alpha)\Delta\theta_{2t}^{t'}). \quad (10)$$

That is, transitioning into a regime with constant model gradients during training can imply LMC because the ensemble and weight-averaged model become near-identical. This also has as an immediate corollary that models with the same $\theta_0$ which train fully within this regime (e.g. those discussed in [JGH18, LXS$^+$19]) will have $t^* = 0$. Note that, when using nonlinear (final) output activation $\sigma(\cdot)$ the post-activation model gradients will generally *not* become constant during training (as we discuss in Sec. 5 for the sigmoid and as was shown theoretically in [LZB20] for general nonlinearities). If, however, the *pre-activation model gradients* become constant during training and the *pre-activation ensemble* – which averages the two model's pre-activation outputs *before* applying $\sigma(\cdot)$ – performs no worse than the individual models (as is also usually the case in practice [JLCvdS24]), then the above also immediately implies LMC for such models.

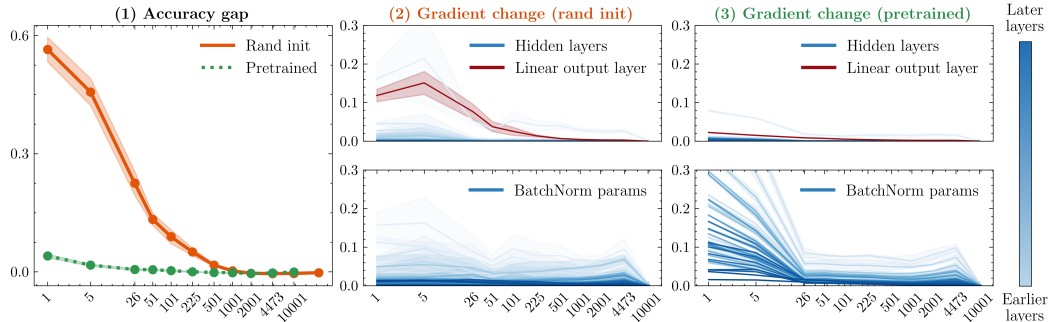

Figure 6: **Linear mode connectivity and gradient changes by** $t'$. (1) Decrease in accuracy when using averaged weights $\alpha\boldsymbol{\theta}_{1T}^{t'} + (1-\alpha)\boldsymbol{\theta}_{2T}^{t'}$ for randomly initialized (orange) and pre-trained ResNet-20 (green). (2) & (3) Changes in model gradients by layer for a randomly initialized (2) and pretrained (3) model.

This suggests a candidate explanation for why LMC emerged at specific points in [FDRC20]. To test this, we replicate their CIFAR-10 experiment using a ResNet-20 in Fig. 6. In addition to plotting the maximal decrease in accuracy when comparing $f_{\alpha\boldsymbol{\theta}_{1T}^{t'}+(1-\alpha)\boldsymbol{\theta}_{2T}^{t'}}(\mathbf{x})$ to the weighted average of the accuracies of the original models as [FDRC20] to measure LMC in (1), we also plot the squared change in (pre-softmax) gradients $(\nabla_{\boldsymbol{\theta}}f_{\boldsymbol{\theta}_{t'+390}}(\mathbf{x}) - \nabla_{\boldsymbol{\theta}}f_{\boldsymbol{\theta}_{t'}}(\mathbf{x}))^2$ over the next epoch (390 batches) after checkpoint $t'$, averaged over the test set and the parameters in each layer in (2). We find that the disappearance of the loss barrier indeed coincides with the time in training when the model gradients become *more stable* across all layers. Most saliently, the appearance of LMC appears to correlate with the stabilization of the gradients of the linear output layer. However, we also continue to observe some changes in other model gradients, which indicates that these models do not train fully linearly.

**Pre-training and weight averaging.** Because weight averaging methods have become increasingly popular when using *pre-trained* instead of randomly initialized models [NSZ20, WIG+22, CVSK22], we are interested in testing whether pre-training may improve mode connectability through stabilizing the model gradients. To test this, we replicate the above experiment with the same architecture pre-trained on the SVHN dataset (in green in Fig. 6(1)). Mimicking findings of [NSZ20], we first find the loss barrier to be substantially lower after pre-training. In Fig. 6(3), we then observe that the gradients in the hidden and final layers indeed change less and stabilize earlier in training than in the randomly initialized model – yet the gradients of the BatchNorm parameters change *more*. Overall, the findings in this section thus highlight that while there may be a connection between gradient stabilization and LMC, it cannot fully explain it – suggesting that further investigation into the phenomenon using this lens, particularly into the role of BatchNorm layers, may be fruitful.

*Takeaway Case Study 3.* Reasoning through the learning process by telescoping out functional updates suggests that averaging model parameters trained from the same checkpoint can be effective if their models' gradients remain stable, however, this cannot fully explain LMC in the setting we consider.

## 5 The Effect of Design Choices on Linearized Functional Updates

The literature on the neural tangent kernel primarily considers plain SGD, while modern deep learning practice typically relies on a range of important modifications to the training process (see e.g. [Pri23, Ch. 6]) – this includes many of the experiments demonstrating surprising deep learning phenomena we examined in Sec. 4. To enable us to use modern optimizers above, we derived their implied linearized functional updates through the weight-averaging representation $\Delta\tilde{f}_t(\mathbf{x}) = \nabla_{\boldsymbol{\theta}}f_{\boldsymbol{\theta}_{t-1}}(\mathbf{x})^\top\Delta\boldsymbol{\theta}_t$, which in turn allows us to define $K_t^T(\cdot,\cdot)$ in Eq. (5) for these modifications using straightforward algebra. As a by-product, we found that this provides us with an interesting and pedagogical formalism to reason about the relative effect of different design choices in neural network training, and elaborate on selected learnings below.

• **Momentum** with scalar hyperparameter $\beta_1$ smoothes weight updates by employing an exponentially weighted average over the previous parameter gradients as $\Delta\boldsymbol{\theta}_t = -\gamma_t\frac{1-\beta_1}{1-\beta_1^t}\sum_{k=1}^t\beta_1^{t-k}\mathbf{T}_k\mathbf{g}_k^\ell$ instead of using the current gradients alone. This implies linearized functional updates

$$\Delta\tilde{f}_t(\mathbf{x}) = -\gamma_t\frac{1-\beta_1}{1-\beta_1^t}\sum_{i\in[n]}(K_t^{\boldsymbol{\theta}}(\mathbf{x},\mathbf{x}_i)g_{it}^\ell + \sum_{k=1}^{t-1}\beta_1^{t-k}K_{t,k}^{\boldsymbol{\theta}}(\mathbf{x},\mathbf{x}_i)g_{ik}^\ell) \quad (11)$$

where $K_{t,k}^{\boldsymbol{\theta}}(\mathbf{x},\mathbf{x}_i) := \frac{\mathbf{1}\{i\in B_k\}}{|B_k|}\nabla_{\boldsymbol{\theta}}f_{\boldsymbol{\theta}_{t-1}}(\mathbf{x})^\top\nabla_{\boldsymbol{\theta}}f_{\boldsymbol{\theta}_{k-1}}(\mathbf{x}_i)$ denotes the cross-temporal tangent kernel. Thus, the functional updates also utilize *previous* loss gradients, where their weight is determined

using an inner product of the model gradient features from different time steps. If $\nabla_{\boldsymbol{\theta}} f_{\boldsymbol{\theta}_t}(\mathbf{x})$ is constant throughout training and we use full-batch GD, then the contribution of each training example $i$ to $\Delta \tilde{f}_t(\mathbf{x})$ reduces to $-\gamma_t K_0^{\boldsymbol{\theta}}(\mathbf{x}, \mathbf{x}_i) \frac{1-\beta_1}{1-\beta_1^t} [\sum_{k=1}^t \beta_1^{t-k} g_{ik}^{\ell}]$, an exponentially weighted moving average over its past loss gradients – making the effect of momentum on functional updates analogous to its effect on updates in parameter space. However, if $\nabla_{\boldsymbol{\theta}} f_{\boldsymbol{\theta}_t}(\mathbf{x})$ changes over time, it is e.g. possible that $K_{k,t}^{\boldsymbol{\theta}}(\mathbf{x}, \mathbf{x}_i)$ has opposite sign from $K_t^{\boldsymbol{\theta}}(\mathbf{x}, \mathbf{x}_i)$ in which case momentum reduces instead of amplifies the effect of a previous $g_{it}^{\ell}$. This is more obvious when re-writing Eq. (11) to collect all terms containing a specific $g_{it}^{\ell}$, leading to $K_t^T(\mathbf{x}, \mathbf{x}_i) = \sum_{k=t}^T \gamma_k \frac{1-\beta_1}{1-\beta_1^k} \beta_1^{k-t} K_{k,t}(\mathbf{x}, \mathbf{x}_i)$ for Eq. (5).

- **Weight decay** with scalar hyperparameter $\lambda$ uses $\Delta \boldsymbol{\theta}_t = -\gamma_t (\mathbf{T}_t \mathbf{g}_t^{\ell} + \lambda \boldsymbol{\theta}_{t-1})$. For constant learning rate $\gamma$ this gives $\boldsymbol{\theta}_t = \boldsymbol{\theta}_0 - \sum_{k=1}^t \gamma (\mathbf{T}_k \mathbf{g}_k^{\ell} + \lambda \boldsymbol{\theta}_{k-1}) = (1-\lambda\gamma)^t \boldsymbol{\theta}_0 - \gamma \sum_{k=1}^t (1-\lambda\gamma)^{t-k} \mathbf{T}_k \mathbf{g}_k^{\ell}$. This then implies linearized functional updates

$$\Delta \tilde{f}_t(\mathbf{x}) = -\gamma \sum_{i \in [n]} (K_t(\mathbf{x}, \mathbf{x}_i) g_{it}^{\ell} - \lambda\gamma \sum_{k=1}^{t-1} (1-\lambda\gamma)^{t-1-k} K_{t,k}(\mathbf{x}, \mathbf{x}_i) g_{ik}^{\ell})$$
$$-\gamma\lambda(1-\lambda\gamma)^{t-1} \nabla_{\boldsymbol{\theta}} f_{\boldsymbol{\theta}_{t-1}}(\mathbf{x})^{\top} \boldsymbol{\theta}_0 \tag{12}$$

For full-batch GD and constant tangent kernels, $-\gamma K_0^{\boldsymbol{\theta}}(\mathbf{x}, \mathbf{x}_i)[g_{it} - \lambda\gamma \sum_{k=1}^{t-1} (1-\lambda\gamma)^{t-1-k} g_{ik}]$ is the contribution of each training example to the functional updates, which effectively decays the previous contributions of this example. Further, comparing the signs in Eq. (12) to Eq. (11) highlights that momentum can *offset* the effect of weight decay on the learned updates in function space (in which case weight decay mainly acts through the term decaying the initial weights $\boldsymbol{\theta}_0$).

- **Adaptive & parameter-dependent learning rates** are another important modification in practice which enable the use of different step-sizes across parameters by dividing $\Delta \boldsymbol{\theta}_t$ *elementwise* by a $p \times 1$ scaling vector $\phi_t$. Most prominently, this is used to adaptively normalize the magnitude of updates (e.g. Adam [KB14] uses $\phi_t = \sqrt{\frac{1-\beta_2}{1-\beta_2^t} \sum_{k=1}^t \beta_2^{t-k} [\mathbf{T}_k \mathbf{g}_k^{\ell}]^2 + \epsilon}$). When combined with plain SGD, this results in kernel $K_t^{\phi}(\mathbf{x}, \mathbf{x}_i) = \frac{\mathbf{1}\{i \in B_t\}}{|B_t|} \nabla_{\boldsymbol{\theta}} f_{\boldsymbol{\theta}_{t-1}}(\mathbf{x})^{\top} \text{diag}(\frac{1}{\phi_t}) \nabla_{\boldsymbol{\theta}} f_{\boldsymbol{\theta}_{t-1}}(\mathbf{x}_i)$. This expression highlights that $\phi_t$ admits an elegant interpretation as *re-scaling the relative influence of features* on the tangent kernel, similar to structured kernels in non-parametric regression [HTF09, Ch. 6.4.1].

- **Architecture design choices** also impact the form of the kernel. One important practical example is whether $f_{\boldsymbol{\theta}}(\mathbf{x})$ applies a non-linear activation function to the output $g_{\boldsymbol{\theta}}(\mathbf{x}) \in \mathbb{R}$ of its final layer. Consider the choice of using the sigmoid $\sigma(z) = \frac{1}{1+e^{-z}}$ for a binary classification problem and recall $\frac{\partial}{\partial z} \sigma(z) = \sigma(z)(1 - \sigma(z)) \in (0, 1/4]$, which is largest where $\sigma(z) = 1/2$ and smallest when $\sigma(z) \to 0 \vee 1$. If $K_t^{\boldsymbol{\theta},g}(\mathbf{x}, \mathbf{x}_i) := \frac{\mathbf{1}\{i \in B_t\}}{|B_t|} \nabla_{\boldsymbol{\theta}} g_{\boldsymbol{\theta}_{t-1}}(\mathbf{x})^{\top} \nabla_{\boldsymbol{\theta}} g_{\boldsymbol{\theta}_{t-1}}(\mathbf{x}_i)$ denotes the tangent kernel of the model without activation, it is easy to see that the tangent kernel of the model $\sigma(g_{\boldsymbol{\theta}_t}(\mathbf{x}))$ is

$$K_t^{\boldsymbol{\theta},\sigma}(\mathbf{x}, \mathbf{x}_i) = \sigma(g_{\boldsymbol{\theta}_t}(\mathbf{x}))(1 - \sigma(g_{\boldsymbol{\theta}_t}(\mathbf{x})))\sigma(g_{\boldsymbol{\theta}_t}(\mathbf{x}_i))(1 - \sigma(g_{\boldsymbol{\theta}_t}(\mathbf{x}_i)))K_t^{\boldsymbol{\theta},g}(\mathbf{x}, \mathbf{x}_i) \tag{13}$$

indicating that $K_t^{\boldsymbol{\theta},\sigma}(\mathbf{x}, \mathbf{x}_i)$ will give relatively higher weight in functional updates to training examples $i$ for which the model is uncertain ($\sigma(g(\mathbf{x}_i)) \approx 1/2$)) and lower weight to examples where the model is certain ($\sigma(g_{\boldsymbol{\theta}_t}(\mathbf{x}_i)) \approx 0 \vee 1$) – *regardless* of whether $\sigma(g_{\boldsymbol{\theta}_t}(\mathbf{x}_i))$ is the correct label. Conversely, Eq. (13) also implies that when comparing the functional updates of $\sigma(g_{\boldsymbol{\theta}}(\mathbf{x}))$ to those of $g_{\boldsymbol{\theta}}(\mathbf{x})$ across inputs $\mathbf{x} \in \mathcal{X}$, updates with $\sigma(\cdot)$ will be relatively larger for $\mathbf{x}$ where the model is uncertain ($\sigma(g_{\boldsymbol{\theta}_t}(\mathbf{x})) \approx 1/2$)). Finally, Eq. (13) also highlights that the (post-activation) tangent kernel of a model with sigmoid activation will generally not be constant in $t$ unless the model predictions $\sigma(g_{\boldsymbol{\theta}_t}(\mathbf{x}))$ do not change.

## 6 Conclusion

This work investigated the utility of a telescoping model for neural network learning, consisting of a sequence of linear approximations, as a tool for understanding several recent deep learning phenomena. By revisiting existing empirical observations, we demonstrated how this perspective provides a lens through which certain surprising behaviors of deep learning can become more intelligible. In each case study, we intentionally restricted ourselves to specific, noteworthy empirical examples which we proceeded to re-examine in greater depth. We believe that there are therefore many interesting opportunities for future research to expand on these initial findings by building upon the ideas we present to investigate such phenomena in more generality, both empirically and theoretically.

**Acknowledgements**

We would like to thank James Bayliss, who first suggested to us to look into explicitly unravelling SGD updates to write trained neural networks as approximate smoothers to study deep double descent after a seminar on our paper [CJvdS23] on non-deep double descent. This suggestion ultimately inspired many investigations far beyond the original double descent context. We are also grateful to anonymous reviewers for helpful comments and suggestions. AC and AJ gratefully acknowledge funding from AstraZeneca and the Cystic Fybrosis Trust, respectively. This work was supported by a G-Research grant, and Azure sponsorship credits granted by Microsoft's AI for Good Research Lab.

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

# Appendix

This appendix is structured as follows: Appendix A presents an extended literature review, Appendix B presents additional theoretical derivations, Appendix C presents an extended discussion of experimental setups and Appendix D presents additional results. The NeurIPS paper checklist is included after the appendices.

## A  Additional literature review

In this section, we present an extended literature review related to the phenomena we consider in Sec. 4.1 and Sec. 4.3.

### A.1  The model complexity-performance relationship (Sec. 4.1)

Classical statistical textbooks convey a well-understood relationship between model complexity – historically captured by a model's parameter count – and prediction error: increasing model complexity is expected to modulate a transition between under- and overfitting regimes, usually represented by a U-shaped error-curve with model complexity on the x-axis in which test error first improves before it worsens as the training data can be fit too well [HT90, Vap95, HTF09]. While this relationship was originally believed to hold for neural networks as well [GBD92], later work provided evidence that – when using parameter counts to measure complexity – this U-shaped relationship no longer holds [NMB$^+$18, Nea19].

**Double descent.**  Instead, the double descent [BHMM19] shape has claimed its place, which postulates that the well-known U-shape holds only in the underparameterized regime where the number of model parameters $p$ is smaller than the number of training examples $n$; once we reach the interpolation threshold $p = n$ at which models have sufficient capacity to fit the training data perfectly, increasing $p$ further into the overparametrized (or: interpolation) regime leads to test error improving again. While the double descent shape itself had been previously observed in linear regression and neural networks in [VCR89, BO96, ASS20, NMB$^+$18, SGd$^+$18] (see also the historical note in [LVM$^+$20]), the seminal paper by [BHMM19] both popularized it as a phenomenon and highlighted that the double descent shape can also occur tree-based methods. In addition to double descent as a function of the number of model parameters, the phenomenon has since been shown to emerge also in e.g. the number of training epochs[NKB$^+$21] and sparsity [HXZQ22]. Optimal regularization has been shown to mitigate double descent [NVKM20].

Due to its surprising and counterintuitive nature, the emergence of the double descent phenomenon sparked a rich theoretical literature attempting to understand it. One strand of this literature has focused on modeling double descent in the number of features in linear regression and has produced precise theoretical analyses for particular data-generating models [BHX20, ASS20, BLLT20, DLM20, HMRT22, SKR$^+$23, CMBK21]. Another strand of work has focused on deriving exact expressions of bias and variance terms as the total number of model parameters is increased in a neural network by taking into account all sources of randomness in model training [NMB$^+$18, AP20, dRBK20, LD21]. A different perspective was presented in [CJvdS23], who highlighted that in the non-deep double descent experiments of [BHMM19], a subtle change in the parameter-increasing mechanism is introduced exactly at the interpolation threshold, which is what causes the second descent. [CJvdS23] also demonstrated that when using a measure of the test-time *effective* parameters used by the model to measure complexity on the x-axes, the double descent shapes observed for linear regression, trees, and boosting fold back into more traditional U-shaped curves. In Sec. 4.1, we show that the telescoping model enables us to discover the same effect also in deep learning.

**Benign overfitting.** Closely related to the double descent phenomenon is *benign overfitting* (e.g. [BMM18, MBB18, BLLT20, CL21, MSA$^+$22, WOBM17, HHLS24]), i.e. the observation that, incompatible with conventional statistical wisdom about overfitting [HTF09], models with perfect training performance can nonetheless generalize well to unseen test examples. In this literature, it is often argued in theoretical studies that overparameterized neural networks generalize well because they are much more well-behaved around unseen test examples than examples seen during training [MSA$^+$22, HHLS24]. In Sec. 4.1 we provide new *empirical* evidence for this by highlighting that there is a difference between $p_{\hat{\mathbf{s}}}^{train}$ and $p_{\hat{\mathbf{s}}}^{test}$.

**Understanding modern model complexity.** Many measures for model complexity capture some form of *capacity* of a hypothesis class, which gives insight into the most complex function that *could be* learned – e.g. raw parameter counts and VC dimensions [BBL03]. The double descent and benign overfitting phenomena prominently highlighted that complexity measures that consider only what *could* be learned and not what *is actually* learned for test examples, would be unlikely to help understand generalization in deep learning [Bel21]. Further, [CJvdS23] highlighted that many other measures for model complexity – so-called measures of effective parameters (or: degrees of freedom) including measures from the literature of smoothers [HT90, Ch. 3.5] as well as measures relying on the model's Hessian [Moo91, Mac91] (which have been considered for use in deep learning in [MBW20]) – were derived in the context of in-sample prediction (where train- and test inputs would be the same) and do thus not allow to distinguish differences in the behavior of learned functions on training examples from new examples. [Cur24] highlight that this difference in setting – the move from in-sample prediction to measuring performance in terms of out-of-sample generalization – is crucial for the emergence of apparently counterintuitive modern machine learning phenomena such as double descent and benign overfitting. For this reason, [CJvdS23] proposed an adapted effective parameter measure for smoothers that can distinguish the two, and highlighted that differentiating between the amount of smoothing performed on train- vs test examples is crucial to understanding double descent in linear regression, trees and gradient boosting. In Sec. 4.1, we show that the telescoping model makes it possible to use [CJvdS23]'s effective parameter measure for neural networks, allowing interesting insight into implied differences in train- and test-time complexity of neural networks.

**Grokking.** Similar to double descent in the number of training epochs as observed in [NKB+21] (where the test error first improves then gets worse and then improves again during training), the *grokking* phenomenon [PBE+22] demonstrated the emergence of another type of unexpected behavior during the training run of a single model. Originally demonstrated on arithmetic tasks, the phenomenon highlights that improvements in test performance can sometimes occur long after perfect training performance has already been achieved. [LMT22] later demonstrated that this can also occur on more standard tasks such as image classification. This phenomenon has attracted much recent attention both because it appears to challenge the common practice of early stopping during training and because it showcases further gaps in our current understanding of learning dynamics. A number of explanations for this phenomenon have been put forward recently: [LKN+22] attribute grokking to delayed learning of representations, [NCL+23] use mechanistic explanations to examine case studies of grokking, [VSK+23] attribute grokking to more efficient circuits being learned later in training, [LMT22] attribute grokking to the effects of weight decay setting in later in training and [TLZ+22] attribute grokking to the use of adaptive optimizers. [KBGP24] highlight that the latter two explanations cannot be the sole reason for grokking by constructing an experiment where grokking occurs as the weight norm grows without the use of adaptive optimizers. Instead, [KBGP24, LJL+24] conjecture that grokking occurs as a model transitions from the lazy regime to a feature learning regime later in training. Finally, [LBBS24] show analytically and experimentally that grokking can also occur in simple linear estimators, and [MOB24] similarly study grokking outside neural networks, including Bayesian models. Our perspective presented in Sec. 4.1 is complementary to these lines of work: we highlight that grokking coincides with the widening of a gap in effective parameters used for training and testing examples and that there is thus a quantifiable benign overfitting effect at play.

## A.2 Weight averaging in deep learning (Sec. 4.3)

Ensembling [Die02], i.e. averaging the *predictions* of multiple independent models, has long established itself as a popular strategy to improve prediction performance over using single individual models. While ensembles have historically been predominantly implemented using weak base learners like trees to form random forests [Bre01], *deep* ensembles [LPB17] – i.e. ensembles of neural networks – have more recently emerged as a popular strategy for improving upon the performance of a single network [LPB17, FHL19]. Interestingly, deep ensembles have been shown to perform well both when averaging the predictions of the underlying models and when averaging the pre-activations of the final network layers [JLCvdS24].

A much more surprising empirical observation made in recent years is that, instead of averaging model predictions as in an ensemble, it is sometimes also possible to average the learned *weights* $\theta_1$ and $\theta_2$ of two trained neural networks and obtain a model that performs well [IPG+18, FDRC20].

This is unexpected because neural networks are *highly nonlinear* functions of their weights, so it is unclear a priori when and why averaging two sets of weights would lead to a sensible model at all. When weight averaging works, it is a much more attractive solution relative to ensembling: an ensemble consisting of $k$ models requires $k \times p$ model parameters, while a weight-averaged model requires only $p$ parameters – making weight-averaged models both more efficient in terms of storage and at inference time. Additionally, weight averaging has interesting applications in federated learning because it could enable the merging of models trained on disjoint datasets. [IPG+18] were the first to demonstrate that weight averaging can work in the context of neural networks by showing that model weights obtained by simple averaging of multiple points along the trajectory of SGD during training – a weight-space version of the method of fast geometric ensembling [GIP+18] – could improve upon using the final solution directly.

**Mode connectivity.** The literature on mode connectivity first empirically demonstrated that there are simple (but nonlinear) paths of nonincreasing loss connecting different final network weights obtained from different random initializations [FB16, DVSH18, GIP+18]. As discussed in the main text, [FDRC20] then demonstrated empirically that two learned sets of weights can sometimes be *linearly* connected by simply interpolating between the learned weights, as long as two models were trained together until some stability point $t^*$. [ABNH23] perform an empirical study investigating which networks and optimization protocols lead to mode connectivity from initialization (i.e. $t^* = 0$) and which modifications ensure $t^* > 0$. As highlighted in Sec. 4.3, our theoretical reasoning indicates that one sufficient condition for linear mode connectivity from initialization is that models stay in a regime in which the model gradients do not change during training. In the context of *task arithmetic*, where parameters from models finetuned on separate tasks are added or subtracted (*not* averaged) to add or remove a skill, [OJFF24] find that pretrained CLIP models that are finetuned on separate tasks and allow to perform task arithmetic *do not* operate in a regime in which gradients are constant.

**Methods that average weights.** Beyond [IPG+18]'s stochastic weight averaging method, which averages weights from checkpoints within a single training run, weight averaging has also recently gained increased popularity in the context of averaging multiple models finetuned from the same pretrained model [NSZ20, WIG+22, CVSK22]: while [NSZ20] showed that multiple models finetuned from the same pretrained model lie in the same loss basin and are linearly mode connectible, the model soups method of [WIG+22] highlighted that simply averaging the weights of multiple models fine-tuned from the same pre-trained parameters with different hyperparameters leads to performance improvements over choosing the best individual fine-tuned model. A number of methods have since been proposed that use weight-averaging of models fine-tuned from the same pretrained model for diverse purposes (e.g. [RKR+22, IWG+22]). Our results in Sec. 4.3 complement the findings of [NSZ20] by investigating whether fine-tuning from a pre-trained model leads to better mode connectivity because the gradients of a pre-trained model remain more stable than those trained from a random initialization.

**Weight averaging after permutation matching.** Most recently, a growing number of papers have investigated whether attempts to merge models through weight-averaging can be improved by first performing some kind of permutation matching that corrects for potential permutation symmetries in neural networks. [ESSN21] conjecture that all solutions learned by SGD are linearly mode connectible once permutation symmetries are corrected for. [SJ20, AHS22, BSM+22] use different methods for permutation matching and find that this improves the quality of weight-averaged models.

# B  Additional theoretical results

## B.1  Derivation of smoother expressions using the telescoping model

Below, we explore how we can use the telescoping model to express a function learned by a neural network as $\tilde{f}_{\boldsymbol{\theta}_t}(\mathbf{x}) = \mathbf{s}_{\boldsymbol{\theta}_t}(\mathbf{x})\mathbf{y} + c_{\boldsymbol{\theta}_t}^0(\mathbf{x})$, where the $1 \times n$ vector $\mathbf{s}_{\boldsymbol{\theta}_t}(\mathbf{x})$ is a function of the kernels $\{K_{t'}^t(\cdot, \cdot)\}_{t' \leq t}$, and the scalar $c_{\boldsymbol{\theta}_t}^0(\mathbf{x})$ is a function of the $\{K_{t'}^t(\cdot, \cdot)\}_{t' \leq t}$ and the networks' initialization $f_{\boldsymbol{\theta}_0}(\cdot)$. Note that, as discussed further in the remark at the end of this section, the kernels $K_{t'}^t(\cdot, \cdot)$ for $t > 1$ are *data-adaptive* as they can change throughout training.

**Vanilla SGD.** Recall that letting $\mathbf{y} = [y_1, \ldots, y_n]^\top$ and $\mathbf{f}_{\boldsymbol{\theta}_t} = [f_{\boldsymbol{\theta}_t}(\mathbf{x}_1), \ldots, f_{\boldsymbol{\theta}_t}(\mathbf{x}_n)]$, the SGD weight update with squared loss $\ell(f(\mathbf{x}), y) = \frac{1}{2}(y - f(\mathbf{x}))^2$, in the special case of single outputs $k = 1$, simplifies to $\Delta\boldsymbol{\theta}_t = \gamma_t \mathbf{T}_t(\mathbf{y} - \mathbf{f}_{\boldsymbol{\theta}_{t-1}})$, where $\mathbf{T}_t$ is the $p \times n$ matrix

$\mathbf{T}_t = [\frac{\mathbf{1}\{1\in B_t\}}{|B_t|}\nabla_{\boldsymbol{\theta}} f_{\boldsymbol{\theta}_{t-1}}(\mathbf{x}_1), \ldots, \frac{\mathbf{1}\{n\in B_t\}}{|B_t|}\nabla_{\boldsymbol{\theta}} f_{\boldsymbol{\theta}_{t-1}}(\mathbf{x}_n)]$. If we assume that the telescoping model holds exactly, this implies functional updates $\Delta \tilde{f}_t(\mathbf{x}) = \gamma_t \nabla_{\boldsymbol{\theta}} f_{\boldsymbol{\theta}_{t-1}}(\mathbf{x})^\top \mathbf{T}_t(\mathbf{y} - \tilde{\mathbf{f}}_{\boldsymbol{\theta}_{t-1}})$. If we could write $\tilde{\mathbf{f}}_{\boldsymbol{\theta}_{t-1}} = \mathbf{S}_{\boldsymbol{\theta}_{t-1}}\mathbf{y} + \mathbf{c}_{\boldsymbol{\theta}_{t-1}}$, then we would have

$$\begin{aligned}
\Delta\tilde{f}_t(\mathbf{x}) &= \gamma_t\nabla_{\boldsymbol{\theta}} f_{\boldsymbol{\theta}_{t-1}}(\mathbf{x})^\top \mathbf{T}_t(\mathbf{y} - (\mathbf{S}_{\boldsymbol{\theta}_{t-1}}\mathbf{y} + \mathbf{c}_{\boldsymbol{\theta}_{t-1}})) \\
&= \gamma_t\nabla_{\boldsymbol{\theta}} f_{\boldsymbol{\theta}_{t-1}}(\mathbf{x})^\top \mathbf{T}_t(\mathbf{I}_n - \mathbf{S}_{\boldsymbol{\theta}_{t-1}})\mathbf{y} - \gamma_t\nabla_{\boldsymbol{\theta}} f_{\boldsymbol{\theta}_{t-1}}(\mathbf{x})^\top \mathbf{T}_t\mathbf{c}_{\boldsymbol{\theta}_{t-1}}
\end{aligned} \tag{14}$$

where $\mathbf{I}_n$ is the $n \times n$ identity matrix. Noting that we must have $\mathbf{c}_{\boldsymbol{\theta}_0} = \mathbf{f}_{\boldsymbol{\theta}_0}$ and $\mathbf{S}_{\boldsymbol{\theta}_0} = \mathbf{0}^{n\times n}$ at initialization, we can recursively substitute Eq. (14) into Eq. (5) which then allows to write the vector of training predictions as

$$\begin{aligned}
\tilde{\mathbf{f}}_{\boldsymbol{\theta}_T} = \underbrace{\left(\sum_{t=1}^{T}\left(\prod_{k=1}^{T-t}(\mathbf{I}_n - \gamma_{t+k}\bar{\mathbf{T}}_{t+k}^\top\mathbf{T}_{t+k})\right)\gamma_t\bar{\mathbf{T}}_t^\top\mathbf{T}_t\right)\mathbf{y}}_{\mathbf{S}_{\boldsymbol{\theta}_T}\mathbf{y}} \\
+ \underbrace{\left(\prod_{k=0}^{T-1}(\mathbf{I}_n - \gamma_{T-k}\bar{\mathbf{T}}_{T-k}^\top\mathbf{T}_{T-k})\right)\mathbf{f}_{\boldsymbol{\theta}_0}}_{\mathbf{c}_{\boldsymbol{\theta}_T}}
\end{aligned} \tag{15}$$

where the $p \times n$ matrix $\bar{\mathbf{T}}_t = [\nabla_{\boldsymbol{\theta}} f_{\boldsymbol{\theta}_{t-1}}(\mathbf{x}_1), \ldots, \nabla_{\boldsymbol{\theta}} f_{\boldsymbol{\theta}_{t-1}}(\mathbf{x}_n)]$ differs from $\mathbf{T}_t$ only in that it includes *all* training examples and is not normalized by batch size. Then note that Eq. (15) is indeed a function of the training labels $\mathbf{y}$, the predictions at initialization $\mathbf{f}_{\boldsymbol{\theta}_0}$ and the model gradients $\{\bar{\mathbf{T}}_t\}_{t=1}^T$ traversed during training (captured in the $n \times n$ matrix $\mathbf{S}_{\boldsymbol{\theta}_T}$ and the $n \times 1$ vector $\mathbf{c}_{\boldsymbol{\theta}_T}$) *alone*. Similarly, we can also write the weight updates (and, by extension, the weights $\boldsymbol{\theta}_T$) using the same quantities, i.e. $\Delta\boldsymbol{\theta}_t = \gamma_t\mathbf{T}_t(\mathbf{I}_n - \mathbf{S}_{\boldsymbol{\theta}_{t-1}})\mathbf{y} - \gamma_t\mathbf{T}_t\mathbf{c}_{\boldsymbol{\theta}_{t-1}}$. By Eq. (5), this also implies that we can write predictions at *arbitrary* test input points as a function of the same quantities:

$$\tilde{f}_{\boldsymbol{\theta}_T}(\mathbf{x}) = \underbrace{\left(\sum_{t=1}^{T}\gamma_t\nabla_{\boldsymbol{\theta}} f_{\boldsymbol{\theta}_{t-1}}(\mathbf{x})^\top\mathbf{T}_t(\mathbf{I}_n - \mathbf{S}_{\boldsymbol{\theta}_{t-1}})\right)\mathbf{y}}_{\mathbf{s}_{\boldsymbol{\theta}_T}(\mathbf{x})\mathbf{y}} + \underbrace{\left(f_{\boldsymbol{\theta}_0}(\mathbf{x}) - \sum_{t=1}^{T}\gamma_t\nabla_{\boldsymbol{\theta}} f_{\boldsymbol{\theta}_{t-1}}(\mathbf{x})^\top\mathbf{T}_t\mathbf{c}_{\boldsymbol{\theta}_{t-1}}\right)}_{c_{\boldsymbol{\theta}_T}(\mathbf{x})}$$

where the matrix $\mathbf{S}_{\boldsymbol{\theta}_{t-1}}$ is as defined in Eq. (15), which indeed has $\mathbf{s}_{\boldsymbol{\theta}_{t-1}}(\mathbf{x}_i)$ as its $i$-th row (and analogously for $\mathbf{c}_{\boldsymbol{\theta}_{t-1}}$).

**General optimization strategies.** Adapting the previous expressions to enable the use of adaptive learning rates is straightforward and requires only inserting $\text{diag}(\frac{1}{\phi_t})\mathbf{T}_t$ into the expression for $\Delta\tilde{f}_t(\mathbf{x})$ instead of $\mathbf{T}_t$ alone; then defining the matrices similarly proceeds by recursively unraveling updates using $\Delta\tilde{f}_t(\mathbf{x}) = \gamma_t\nabla_{\boldsymbol{\theta}} f_{\boldsymbol{\theta}_{t-1}}(\mathbf{x})^\top\text{diag}(\frac{1}{\phi_t})\mathbf{T}_t(\mathbf{y} - \tilde{\mathbf{f}}_{\boldsymbol{\theta}_{t-1}})$. Both momentum and weight decay lead to somewhat more tedious updates and necessitate the introduction of additional notation. Let $\Delta\mathbf{s}_t(\mathbf{x}) = \mathbf{s}_{\boldsymbol{\theta}_t}(\mathbf{x}) - \mathbf{s}_{\boldsymbol{\theta}_{t-1}}(\mathbf{x})$, with $\mathbf{s}_{\boldsymbol{\theta}_0}(\mathbf{x}) = \mathbf{0}^{1\times n}$ and $\Delta c_t(\mathbf{x}) = c_{\boldsymbol{\theta}_t}(\mathbf{x}) - c_{\boldsymbol{\theta}_{t-1}}(\mathbf{x})$, with $c_{\boldsymbol{\theta}_0}(\mathbf{x}) = f_{\boldsymbol{\theta}_0}(\mathbf{x})$, so that $\mathbf{s}_{\boldsymbol{\theta}_T}(\mathbf{x}) = \sum_{t=1}^{T}\Delta\mathbf{s}_t(\mathbf{x})$ and $\mathbf{c}_{\boldsymbol{\theta}_T}(\mathbf{x}) = f_{\boldsymbol{\theta}_0}(\mathbf{x}) + \sum_{t=1}^{T}\Delta c_t(\mathbf{x})$. Further, we can write

$$\Delta\tilde{f}_t(\mathbf{x}) = \Delta\mathbf{s}_t(\mathbf{x})\mathbf{y} + \mathbf{c}_t(\mathbf{x}) = \gamma_t\nabla_{\boldsymbol{\theta}} f_{\boldsymbol{\theta}_{t-1}}(\mathbf{x})^\top\mathbf{U}_t^S\mathbf{y} + \gamma_t\nabla_{\boldsymbol{\theta}} f_{\boldsymbol{\theta}_{t-1}}(\mathbf{x})^\top\mathbf{U}_t^C \tag{16}$$

which means that to derive $\mathbf{s}_{\boldsymbol{\theta}_t}(\mathbf{x})$ for each $t$, we can use the weight update formulas to define the $p \times n$ update matrix $\mathbf{U}_t^S$ and the $p \times 1$ update vector $\mathbf{U}_t^C$ that can then be used to compute $\Delta\mathbf{s}_t(\mathbf{x})$ as $\gamma_t\nabla_{\boldsymbol{\theta}} f_{\boldsymbol{\theta}_{t-1}}(\mathbf{x})^\top\mathbf{U}_t^S$ and $\Delta\mathbf{c}_t(\mathbf{x})$ as $\gamma_t\nabla_{\boldsymbol{\theta}} f_{\boldsymbol{\theta}_{t-1}}(\mathbf{x})^\top\mathbf{U}_t^C$. For vanilla SGD,

$$\mathbf{U}_t^S = \mathbf{T}_t(\mathbf{I}_n - \mathbf{S}_{\boldsymbol{\theta}_{t-1}}) \text{ and } \mathbf{U}_t^C = -\mathbf{T}_t\mathbf{c}_{\boldsymbol{\theta}_{t-1}} \tag{17}$$

while SGD with only adaptive learning rates uses

$$\mathbf{U}_t^S = \text{diag}(\frac{1}{\phi_t})\mathbf{T}_t(\mathbf{I}_n - \mathbf{S}_{\boldsymbol{\theta}_{t-1}}) \text{ and } \mathbf{U}_t^C = -\text{diag}(\frac{1}{\phi_t})\mathbf{T}_t\mathbf{c}_{\boldsymbol{\theta}_{t-1}} \tag{18}$$

Momentum, without other modifications, uses $\mathbf{U}_t^S = \frac{1}{1-\beta_1^t}\tilde{\mathbf{U}}_t^S$ and $\mathbf{C}_t^S = \frac{1}{1-\beta_1^t}\tilde{\mathbf{U}}_t^C$, where

$$\tilde{\mathbf{U}}_t^S = (1-\beta_1)\mathbf{T}_t(\mathbf{I}_n - \mathbf{S}_{\boldsymbol{\theta}_{t-1}}) + \beta_1\tilde{\mathbf{U}}_{t-1}^S \text{ and } \tilde{\mathbf{U}}_t^C = -((1-\beta_1)\mathbf{T}_t\mathbf{c}_{\boldsymbol{\theta}_{t-1}} + \beta_1\tilde{\mathbf{U}}_{t-1}^C) \tag{19}$$

with $\tilde{\mathbf{U}}_0^S = \mathbf{0}^{p \times n}$ and $\tilde{\mathbf{U}}_0^S = \mathbf{0}^{p \times 1}$.

Weight decay, without other modifications, uses

$$\mathbf{U}_t^S = \mathbf{T}_t(\mathbf{I}_n - \mathbf{S}_{\boldsymbol{\theta}_{t-1}} + \lambda \mathbf{D}_t^S) \text{ and } \mathbf{U}_t^C = -\mathbf{T}_t(\mathbf{c}_{\boldsymbol{\theta}_{t-1}} + \lambda \mathbf{D}_t^C) \tag{20}$$

where $\mathbf{D}_t^S = \gamma_{t-1}\mathbf{U}_{t-1}^S + (1 - \lambda\gamma_{t-1})\mathbf{D}_{t-1}^S$ and $\mathbf{D}_t^C = \gamma_{t-1}\mathbf{U}_{t-1}^C + (1 - \lambda\gamma_{t-1})\mathbf{D}_{t-1}^C$ with $\mathbf{D}_0^S = \mathbf{0}^{p \times n}$ and $\mathbf{D}_0^C = \boldsymbol{\theta}_0$.

Putting all together leads to AdamW [LH17] (which decouples weight decay and momentum, so that weight decay does not enter the momentum term), which uses

$$\mathbf{U}_t^S = \text{diag}(\frac{1}{\boldsymbol{\phi}_t})\frac{1}{1 - \beta_1^t}\tilde{\mathbf{U}}_t^S + \lambda\mathbf{T}_t\mathbf{D}_t^S \text{ and } \mathbf{C}_t^S = \frac{1}{1 - \beta_1^t}\text{diag}(\frac{1}{\boldsymbol{\phi}_t})\tilde{\mathbf{U}}_t^C + \lambda\mathbf{T}_t\mathbf{D}_t^C \tag{21}$$

where all terms are as in Eq. (19) and Eq. (20).

*Remark:* Writing $\tilde{\mathbf{f}}_{\boldsymbol{\theta}_T} = \mathbf{S}_{\boldsymbol{\theta}_T}\mathbf{y} + \mathbf{c}_{\boldsymbol{\theta}_T}$ is reminiscent of a *smoother* as used in the statistics literature [HT90]. Prototypical smoothers issue predictions $\hat{\mathbf{y}} = \mathbf{S}\mathbf{y}$ – which include k-Nearest Neighbor regressors, kernel smoother, and (local) linear regression as prominent members –, and are usually *linear smoothers* because $\mathbf{S}$ does not depend on $\mathbf{y}$. The smoother implied by the telescoping model is *not* necessarily a linear smoother because $\mathbf{S}_{\boldsymbol{\theta}_T}$ can depend on $\mathbf{y}$ through changes in gradients during training, making $\tilde{\mathbf{f}}_{\boldsymbol{\theta}_T}$ an *adaptive* smoother. This adaptivity in the implied smoother is similar to trees as recently studied in [CJvdS23, CJvdS24]. In this context, effective parameters as measured by $p_s^0$ can be interpreted as measuring how non-uniform and extreme the learned smoother weights are when issuing predictions for specific inputs [CJvdS23].

### B.2 Comparing predictions of ensemble and weight-averaged model after train-time transition into a constant-gradient regime

Here, we compare the predictions of the weight-averaged model $f_{\alpha\boldsymbol{\theta}_{1T}^{t'} + (1-\alpha)\boldsymbol{\theta}_{2T}^{t'}}(\mathbf{x})$ to the ensemble $\bar{f}^\alpha(\mathbf{x}) = \alpha f_{\alpha\boldsymbol{\theta}_{1T}^{t'}}(\mathbf{x}) + (1 - \alpha)f_{\alpha\boldsymbol{\theta}_{2T}^{t'}}(\mathbf{x})$ if the models transition into a lazy regime at time $t^* \leq t'$.

We begin by noting that the assumption that the gradients no longer change after $t^*$ (i.e. $\nabla_{\boldsymbol{\theta}}f_{\boldsymbol{\theta}_{jt}^{t'}}(\cdot) \approx \nabla_{\boldsymbol{\theta}}f_{\boldsymbol{\theta}_{t^*}}(\cdot)$ for all $t \geq t^*$) implies that the rate of change of $\nabla_{\boldsymbol{\theta}}f_{\boldsymbol{\theta}_{t^*}}(\mathbf{x})$ in the direction of the weight updates must be approximately $\mathbf{0}$. That is, $\nabla_{\boldsymbol{\theta}}^2 f_{\boldsymbol{\theta}_{t^*}}(\mathbf{x})(\boldsymbol{\theta} - \boldsymbol{\theta}_{t^*}) \approx \mathbf{0}$ for all $\boldsymbol{\theta} \in \Theta_j^{stable}$, or equivalently all weight changes in each $\Theta_j^{stable}$ are in directions that are in the null-space of the Hessian (or in directions corresponding to diminishingly small eigenvalues). To avoid clutter in notation, we use splitting point $t' = t^*$ below, but note that the same arguments hold for $t' > t^*$.

First, we now consider rewriting the predictions of the ensemble, and note that we can now write the *second-order* Taylor approximation of each model $f_{\boldsymbol{\theta}_{jT}^{t*}}(\mathbf{x})$ around $\boldsymbol{\theta}_{t^*}$ as

$$f_{\boldsymbol{\theta}_{jT}^{t*}}(\mathbf{x}) = f_{\boldsymbol{\theta}_{t*}}(\mathbf{x}) + \nabla_{\boldsymbol{\theta}}f_{\boldsymbol{\theta}_{t*}}(\mathbf{x})^\top \sum_{t=t^*+1}^{T} \Delta\boldsymbol{\theta}_{jt}^{t*} + \frac{1}{2}\underbrace{\left[\sum_{t=t^*+1}^{T}\Delta\boldsymbol{\theta}_{jt}^{t*}\right]^\top \nabla_{\boldsymbol{\theta}}^2 f_{\boldsymbol{\theta}_{t*}}(\mathbf{x})\left[\sum_{t=t^*+1}^{T}\Delta\boldsymbol{\theta}_{jt}^{t*}\right]}_{\approx 0}$$

$$+ R_2(\sum_{t=t^*+1}^{T}\Delta\boldsymbol{\theta}_{jt}^{t*})$$

$$\approx f_{\boldsymbol{\theta}_{t*}}(\mathbf{x}) + \nabla_{\boldsymbol{\theta}}f_{\boldsymbol{\theta}_{t*}}(\mathbf{x})^\top \sum_{t=t^*+1}^{T}\Delta\boldsymbol{\theta}_{jt}^{t*} + R_2(\sum_{t=t^*+1}^{T}\Delta\boldsymbol{\theta}_{jt}^{t*})$$

where $R_2(\sum_{t=t^*+1}^{T}\Delta\boldsymbol{\theta}_{jt}^{t*})$ contains remainders of order 3 and above. Then the prediction of the ensemble can be written as

$$\bar{f}^\alpha(\mathbf{x}) \approx f_{\boldsymbol{\theta}_{t*}}(\mathbf{x}) + f_{\boldsymbol{\theta}_{t*}}(\mathbf{x})^\top \sum_{t=t^*+1}^{T}(\alpha\Delta\boldsymbol{\theta}_{1t}^{t*} + (1-\alpha)\Delta\boldsymbol{\theta}_{2t}^{t*})$$

$$+ \alpha R_2(\sum_{t=t^*+1}^{T}\Delta\boldsymbol{\theta}_{1t}^{t*}) + (1-\alpha)R_2(\sum_{t=t^*+1}^{T}\Delta\boldsymbol{\theta}_{2t}^{t*})) \tag{22}$$

Now consider the weight-averaged model $f_{\alpha\boldsymbol{\theta}_{1T}^{t'}+(1-\alpha)\boldsymbol{\theta}_{2T}^{t'}}(\mathbf{x})$. Note that we can always write $\boldsymbol{\theta}_{jT}^{t^*} = \boldsymbol{\theta}_0 + \sum_{t=1}^{T}\Delta\boldsymbol{\theta}_{jt}^{t^*} = \boldsymbol{\theta}_{t^*} + \sum_{t=t^*+1}^{T}\Delta\boldsymbol{\theta}_{jt}^{t^*}$ and thus $\alpha\boldsymbol{\theta}_{1T}^{t^*} + (1-\alpha)\boldsymbol{\theta}_{2T}^{t^*} = \boldsymbol{\theta}_{t^*} + \sum_{t=t^*+1}^{T}\left(\alpha\Delta\boldsymbol{\theta}_{1t}^{t^*} + (1-\alpha)\Delta\boldsymbol{\theta}_{2t}^{t^*}\right)$. Further, because $\nabla_{\boldsymbol{\theta}}^2 f_{\boldsymbol{\theta}_{t^*}}(\mathbf{x})\sum_{t=t^*+1}^{T}\Delta\boldsymbol{\theta}_{tj}^{t^*} \approx \mathbf{0}$ for each $j \in \{0,1\}$, we also have that

$$\nabla_{\boldsymbol{\theta}}^2 f_{\boldsymbol{\theta}_{t^*}}(\mathbf{x})\left(\sum_{t=t^*+1}^{T}\alpha\Delta\boldsymbol{\theta}_{t1} + (1-\alpha)\Delta\boldsymbol{\theta}_{t2}\right) \approx \alpha\mathbf{0} + (1-\alpha)\mathbf{0} = \mathbf{0} \tag{23}$$

Then, the second-order Taylor approximation of $f_{\alpha\boldsymbol{\theta}_{1T}^{t'}+(1-\alpha)\boldsymbol{\theta}_{2T}^{t'}}(\mathbf{x})$ around $\boldsymbol{\theta}_{t^*}$ gives

$$\begin{aligned}
f_{\alpha\boldsymbol{\theta}_{1T}^{t'}+(1-\alpha)\boldsymbol{\theta}_{2T}^{t'}}(\mathbf{x}) \approx f_{\boldsymbol{\theta}_{t^*}}(\mathbf{x}) + \nabla_{\boldsymbol{\theta}}f_{\boldsymbol{\theta}_{t^*}}(\mathbf{x})^{\top}&\sum_{t=t^*+1}^{T}(\alpha\Delta\boldsymbol{\theta}_{t1} + (1-\alpha)\Delta\boldsymbol{\theta}_{t2}) \\
&+R_2(\sum_{t=t^*+1}^{T}\Delta\boldsymbol{\theta}_{t1} + (1-\alpha)\Delta\boldsymbol{\theta}_{t2})
\end{aligned} \tag{24}$$

Thus, $f_{\alpha\boldsymbol{\theta}_{1T}^{t'}+(1-\alpha)\boldsymbol{\theta}_{2T}^{t'}}(\mathbf{x}) \approx \bar{f}^{\alpha}(\mathbf{x})$ up to remainder terms of third order and above.

## C   Additional Experimental details

In this section, we provide a complete description of the experimental details throughout this work. Code is provided at `https://github.com/alanjeffares/telescoping-lens`. Each section also reports their respective required compute which was performed on either Azure VMs powered by $4 \times$ NVIDIA A100 GPUs or an NVIDIA RTX A4000 GPU.

### C.1   Case study 1 (Sec. 4.1) and approximation quality experiment (Sec. 3, Fig. 2)

**Double descent experiments.**   In Fig. 3, we replicate [BHMM19, Sec. S.3.3]'s only binary classification experiment which used fully connected ReLU networks with a single hidden layer trained using the squared loss, *without* sigmoid activation, on cat and dog images from CIFAR-10 [KH+09]. Like [BHMM19], we grayscale and downsize images to $d = 8 \times 8$ format and use $n = 1000$ training examples and use SGD with momentum $\beta_1 = 0.95$. We use batch size 100 (resulting in $B = 10$ batches), learning rate $\gamma = 0.0025$, and test on $n_{test} = 1000$ held out examples. We train for up to $e = 30000$ epochs, but stop when training accuracy reaches $100\%$ or when the training squared loss does not improve by more than $10^{-4}$ for 500 consecutive epochs (the former strategy was also employed in [BHMM19], we additionally employ the latter to detect converged networks). We report results using $\{1, 2, 5, 7, 10, 15, 20, 25, 30, 35, 40, 45, 50, 55, 70, 85, 100, 200, 500, 1000, 2000, 5000\}$ hidden units. We repeat the experiment for 4 random seeds and report mean and standard errors in all figures.

In Appendix D.2, we additionally repeat this experiment with the same hyperparameters using MNIST images [LBBH98]. To create a binary classification task, we similarly train the model to distinguish 3-vs-5 from $n = 1000$ images downsampled to $d = 8 \times 8$ format and test on 1000 examples. Likely because the task is very simple, we observe no deterioration in test error in this setting for any hidden size (see Fig. 9). Because [NKB+21] found that double descent can be more apparent in the presence of label noise, we repeat this experiment while adding $20\%$ label noise to the training data, in which case the double descent shape in test error indeed emerges. As above, we repeat both experiments for 4 random seeds and report mean and standard errors in all figures.

Further, in Appendix D.2 we additionally utilize the MNIST-1D dataset [GK24] which was proposed recently as a sandbox for investigating empirical deep learning phenomena. We replicate a binary classification version of their MLP double descent experiment with added 15% label noise from [GK24] (which was itself adapted from the textbook [Pri23]). We select only examples with label 0 and 1, and train fully connected neural networks with a single hidden layer with batch size 100, learning rate $\gamma = 0.01$ for 500 epochs, considering models with $[1, 2, 3, 5, 10, 20, 30, 40, 50, 70, 100, 200, 300, 400]$ hidden units.

Compute: We train `num_settings` $\times$ `num_hidden_sizes` $\times$ `num_seeds` ($\approx 4 \times 22 \times 4 = 352$) models for up to $T = B \times e = 300000$ gradient steps. Training times, which included all gradient

computations to create the telescoping approximation, depended on the dataset and hidden sizes, but completing a single seed for all hidden sizes for one setting took an average of 36 hours.

**Grokking experiments.** In panel (1) of Fig. 4, we replicate the polynomial regression experiment from [KBGP24, Sec. 5] exactly. [KBGP24] use a neural network with a single hidden layer, using custom nonlinearities, of width $n_h = 500$ in which the weights of the final layer are fixed, that is they use

$$f_{\boldsymbol{\theta}}(\mathbf{x}) = \frac{1}{n_h} \sum_{j=1}^{n_h} \phi(\boldsymbol{\theta}_j^\top \mathbf{x}) \text{ where } \phi(h) = h + \frac{\epsilon}{2} h^2 \tag{25}$$

Inputs $x \in R^d$ are sampled from an isotropic Gaussian with variance $\frac{1}{d}$ and targets $y$ are generated as $y(\mathbf{x}) = \frac{1}{2}(\boldsymbol{\beta}^\top \mathbf{x})^2$. In this setup, $\epsilon$ used in the activation function of the network controls how easy it is to fit the outcome function (the larger $\epsilon$, the better aligned it is for the task at hand), which in turn controls whether grokking appears. In the main text, we present results using $\epsilon = .2$; in Appendix D.2 we additionally present results using $\epsilon = .05$ and $\epsilon = 0.5$. Like [KBGP24], we use $d = 100$, $n_{train} = 550$, $n_{test} = 500$, initialize all weights using standard normals, and train using full-batch gradient descent with $\gamma = B = 500$ on the squared loss. We repeat the experiment for 5 random seeds and report mean and standard errors in all figures.

In panel (2) of Fig. 4, we report an adapted version of [LMT22]'s experiment reporting grokking on MNIST data. To enable the use of our model, we once more consider the binary classification task 3-vs-5 from $n = 1000$ images downsampled to $d = 8 \times 8$ features and test on 1000 held-out examples. Like [LMT22], we use a 3-layer fully connected ReLU network trained with squared loss (*without* sigmoid activation) and larger than usual initialization by using $\alpha \boldsymbol{\theta}_0$ instead of the default initialization $\boldsymbol{\theta}_0$. We report $\alpha = 6$ in the main text and include results with $\alpha = 5$ and $\alpha = 7$ in Appendix D.2. Like [LMT22] we use the AdamW optimizer [LH17] with batches of size 200, $\beta_1 = .9$ and $\beta_2 = .99$, and use weight decay $\lambda = .1$. While [LMT22] use learning rate $10^{-3}$, we need to reduce this by factor 10 to $\gamma = 10^{-4}$ and additionally use linear learning rate warmup over the first 100 batches to ensure that weight updates are small enough to ensure the quality of the telescoping approximation; this is particularly critical because of the large initialization which otherwise results in instability in the approximation early in training. Panel (C) of Fig. 4 uses an identical setup but lets $\alpha = 1$ (i.e. standard initialization) and additionally applies a sigmoid to the output of the network. We repeat these experiments for 4 random seeds and report mean and standard errors in all figures.

Compute: Replicating [KBGP24]'s experiments required training `num_settings` × `num_seeds` ($3 \times 5 = 15$) models for $T = 100,000$ gradient steps. Each training run including all gradient computations took less than 1 hour to complete. Replicating [LMT22]'s experiments required training `num_settings` × `num_seeds` ($3 \times 4 = 12$) for $T = 100,000$ gradient steps. Each training run including all gradient computations took around 5 hours to complete. The MNIST experiments with standard initialization required training `num_settings` × `num_seeds` ($2 \times 4 = 8$) for $T = 1000$ gradient steps, these took no more than 2 hours to complete in total.

**Approximation quality experiment (Fig. 2)** The approximation quality experiment uses the identical MNIST setup, training process and architecture as in the grokking experiments (differing only in that we use standard initialization $\alpha$ and no learning rate warmup). In addition to the vanilla SGD and AdamW experiments presented in the main text, we present additional settings – using momentum alone, weight decay alone and using sigmoid activation – in Appendix D.1. In particular, we use the following hyperparameter settings for the different panels:

- *"SGD":* $\lambda = 0$, $\beta_1 = 0$, no sigmoid.
- *"AdamW":* $\lambda = 0.1$, $\beta_1 = 0.9$, $\beta_2 = .99$, no sigmoid.
- *"SGD + Momentum":* $\lambda = 0$, $\beta_1 = 0.9$, no sigmoid.
- *"SGD + Weight decay":* $\lambda = 0.1$, $\beta_1 = 0$, no sigmoid.
- *"SGD + $\sigma(\cdot)$":* $\lambda = 0$, $\beta_1 = 0$, with sigmoid activation.

We repeat the experiment for 4 random seeds and report mean and standard errors in all figures.

Compute: Creating Fig. 7 required training `num_settings`$\times$`num_seeds` ($5 \times 4 = 20$) for $T = 5,000$ gradient steps. Each training run including all gradient computations took approximately 15 minutes to complete.

## C.2    Case study 2 (Sec. 4.2)

In Figs. 5 and 14 we provide results on tabular benchmark datasets from [GOV22]. We select four datasets with $> 20,000$ examples (`houses`, `superconduct`, `california`, `house_sales`) to ensure there is sufficient hold-out data for evaluation across irregularity proportions. We apply standard preprocessing including log transformations of skewed features and target rescaling. As discussed in the main text, irregular examples are defined by first projecting each (normalized) dataset's input features onto its first principal component and then calculating each example's absolute distance to the empirical median in this space. We note that several recent works have discussed metrics of an examples irregularity or "hardness" (e.g. [KAF+24, SIvdS23]) finding the choice of metric to be highly context-dependent. Therefore we select a principal component prototypicality approach based on its simplicity and transparency. The top $K$ irregular examples are removed from the data (these form the "irregular examples at test-time") and the remainder (the "regular examples") is split into training and testing. We then construct test datasets containing 4000 examples, constructed from a mixture of standard test examples and irregular examples according to each proportion $p$.

We train both a standard neural network (while computing its telescoping approximation as described in Eq. (5)) and a gradient boosted tree model (using [PVG+11]) on the training data. We select hyperparameters by further splitting the training data to obtain a validation set of size 2000 and applying a random search consisting of 25 runs. We use the search spaces suggested in [GOV22]. Specifically, for GBTs we consider `learning_rate` $\in \text{LogNormal}[\log(0.01), \log(10)]$, `num_estimators` $\in \text{LogUniformInt}[10.5, 1000.5]$, and `max_depth` $\in [\text{None}, 2, 3, 4, 5]$ with respective probabilities $[0.1, 0.1, 0.6, 0.1, 0.1]$. For the neural network, we consider `learning_rate` $\in \text{LogUniform}[1e-5, 1e-2]$ and set `batch_size` $= 128$, `num_layers` $= 3$, and `hidden_dim` $= 64$ with ReLU activations throughout. Each model is then trained on the full training set with its optimal parameters and is evaluated on each of test sets corresponding to the various proportions of irregular examples. All models are trained and evaluated for 4 random seeds and we report the mean and a standard error in our results.

As discussed in the main text, we report how the relative relative mean squared error of neural network and GBT (measured as $\frac{MSE_{NN}^p - MSE_{GBT}^p}{MSE_{NN}^0 - MSE_{GBT}^0}$) changes as the proportion $p$ of irregular examples increases and relate this to changes in $\frac{\frac{1}{T}\sum_{t=1}^{T} \max_{j \in \mathcal{I}_{test}^p} ||\mathbf{k}_t(x_j)||}{\frac{1}{T}\sum_{t=1}^{T} \max_{i \in \mathcal{I}_{train}} ||\mathbf{k}_t(\mathbf{x}_i)||}$, which measures how the kernels behave at their extreme during testing relative to the maximum of the equivalent values measured for the training examples such that the test values can be interpreted relative to the kernel at train time (i.e. values $> 1$ can be interpreted as being larger than the largest value observed across the entire training set).

Compute: The hyperparameter search results in `num_searches` $\times$ `num_datasets` $\times$ `num_models` ($25 \times 4 \times 2 = 200$) training runs and evaluations. Then the main experiment requires `num_seeds` $\times$ `num_datasets` $\times$ `num_models` ($4 \times 4 \times 2 = 32$) training runs and `num_seeds` $\times$ `num_datasets` $\times$ `num_models` $\times$ `num_proportions` ($4 \times 4 \times 2 \times 5 = 160$) evaluations. This results in a total of 232 training runs and 360 evaluations. Individual training and evaluation times depend on the model and dataset but generally require $< 1$ hour.

## C.3    Case study 3 (Sec. 4.3)

In Fig. 6 we follow the experimental setup described in [FDRC20]. Specifically, for each model we train for a total of 63,000 iterations over batches of size 128 with stochastic gradient descent. At a predetermined set of checkpoints ($t' \in [0, 4, 25, 50, 100, 224, 500, 1000, 2000, 4472, 10000, 25100]$) we create two copies of the current state of the network and train until completion with different batch orderings, where linear mode connectivity measurements are calculated. This process sometimes also referred to as *spawning* [FDP+20] and is repeated for 3 seeds at each $t'$. The entire process is repeated for 3 seeds resulting in a total of $3 \times 3 = 9$ total values over which we report the mean and a standard error. Momentum is set to 0.9 and a stepwise learning rate is applied beginning at 0.1 and decreasing by a factor of 10 at iterations 32,000 and 48,000. For the ResNet-20 architecture [HZRS16], we use

an implementation from [Ide]. Experiments are conducted on CIFAR-10 [KH+09] where the inputs are normalized with random crops and random horizontal flips used as data augmentations.

Pretraining of the finetuned model model is performed on the SVHN dataset [NWC+11] which is also an image classification task with identically shaped input and output dimensions as CIFAR-10. We use a training setup similar to that of the CIFAR-10 model but set the number of training iterations to 30,000 and perform the stepwise decrease in learning rate at iterations 15,000 and 25,000 decaying by a factor of 5. Three models are trained following this protocol which achieve validation accuracy of 95.5%, 95.5%, and 95.4% on SVHN. We then repeat the CIFAR-10 training protocol for finetuning but parameterize the three initialization with the respective pretrained weights rather than random initialization. We also find that a shorter finetuning period is sufficient and therefore finetune for 12,800 steps with the learning rate decaying by a factor of 5 at steps 6,400 and 9,600.

Also following the protocol of [FDRC20], for each pair of trained spawned networks $(f_{\boldsymbol{\theta}_1} \& f_{\boldsymbol{\theta}_2})$ we consider interpolating their losses (i.e. $\ell_\alpha^{\text{avg}} := \alpha \cdot \ell(f_{\boldsymbol{\theta}_1}(\mathbf{x}), y) + (1-\alpha) \cdot \ell(f_{\boldsymbol{\theta}_2}(\mathbf{x}), y)$) and parameters (i.e. $\ell_\alpha^{\text{lmc}} := \ell(f_{\boldsymbol{\theta}^{\text{lmc}}}(\mathbf{x}), y)$ where $\boldsymbol{\theta}^{\text{lmc}} = \alpha\boldsymbol{\theta}_1 + (1-\alpha)\boldsymbol{\theta}_2$) for 30 equally spaced values of $\alpha \in [0, 1]$. In the upper panel of Fig. 6 we plot the accuracy gap at each checkpoint $t'$ (i.e. the point from which two identical copies of the model are made and independently trained to completion) which is simply defined as the average final validation accuracy of the two individual child models minus the final validation accuracy of the weight averaged version of these two child models. Beyond the original experiment, we also wish to evaluate how the gradients $\nabla f_{\boldsymbol{\theta}_t}(\cdot)$ evolve throughout training. Therefore, in panels (2) and (3) Fig. 6, at each checkpoint $t'$ we also measure the mean squared change in (pre-softmax) gradients $\left(\nabla_{\boldsymbol{\theta}} f_{\boldsymbol{\theta}_{t'+390}}(\mathbf{x}) - \nabla_{\boldsymbol{\theta}} f_{\boldsymbol{\theta}_{t'}}(\mathbf{x})\right)^2$ between the current iteration $t'$ and those at the next epoch $t' + 390$, averaged over a set of $n = 256$ test examples and the parameters in each layer.

Compute: We train `num_outer_seeds` × `num_inner_seeds` × `num_child_models` × `num_checkpoints` ($3 \times 3 \times 2 \times 12 = 216$) networks for the randomly initialized model. For the finetuned model this results in $3 \times 3 \times 2 \times 10 = 180$ training runs. Additionally, we require the pertaining of the 3 base models on SVHN. Combined this results in a total of $216 + 180 + 3 = 399$ training runs. Training each ResNet-20 on CIFAR-10 required <1 hour including additional gradient computations.

### C.4 Data licenses

All image experiments are performed on CIFAR-10 [KH+09], MNIST [LBBH98], MNIST1D [GK24], or SVHN [NWC+11]. Tabular experiments are run on `houses`, `superconduct`, `california`, and `house_sales` from OpenML [VvRBT13] as described in [GOV22]. CIFAR-10 is released with an MIT license. MNIST is released with a Creative Commons Attribution-Share Alike 3.0 license. MNIST1D is released with an Apache-2.0 license. SVHN is released with a CC0:Public Domain license. OpenML datasets are released with a 3-Clause BSD License. All the datasets used in this work are publicly available.

## D  Additional results

### D.1  Additional results on approximation quality (supplementing Fig. 2)

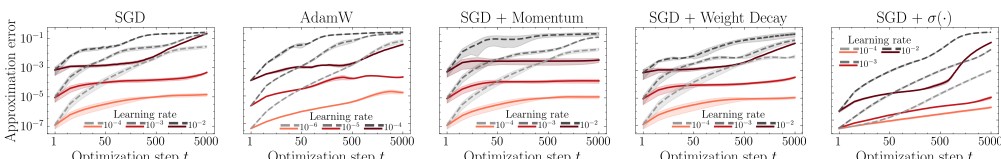

Figure 7: **Approximation error** of the telescoping ($\tilde{f}_{\boldsymbol{\theta}_t}(\mathbf{x})$, red) and the model linearized around the initialization ($f_{\boldsymbol{\theta}_t}^{lin}(\mathbf{x})$, gray) by optimization step for different optimization strategies and other design choices. Iteratively telescoping out the updates using $\tilde{f}_{\boldsymbol{\theta}_t}(\mathbf{x})$ improves upon the lazy approximation around the initialization by orders of magnitude.

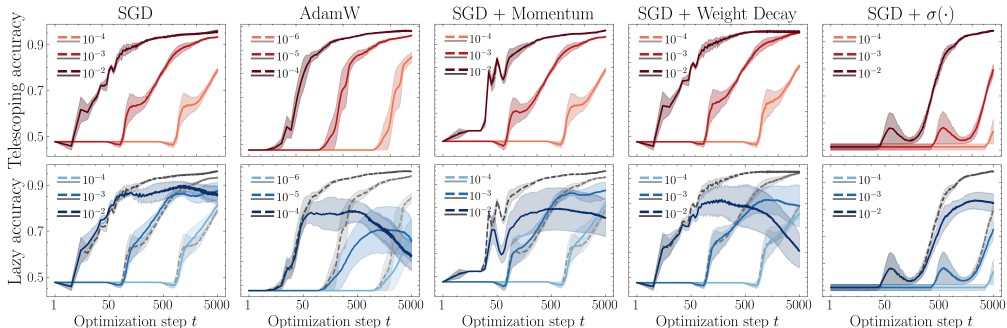

Figure 8: **Test accuracy** of the telescoping ($\tilde{f}_{\boldsymbol{\theta}_t}(\mathbf{x})$, red, top row) and the model linearized around the initialization ($f_{\boldsymbol{\theta}_t}^{lin}(\mathbf{x})$, blue, bottom row) against accuracy of the actual neural network (gray) by optimization step for different optimization strategies and other design choices. While the telescoping model visibly matches the accuracy of the actual neural network, the linear approximation around the initialization leads to substantial differences in accuracy later in training.

In Fig. 7, we present results investigating the evolution of approximation errors of the telescoping and linear approximation around the initialization during training using additional configurations compared to the results presented in Fig. 2 in the main text (replicated in the first two columns of Fig. 7). We observe the same trends as in the main text, where the telescoping approximation matches the predictions by the neural network by orders of magnitudes better than the linear approximation around the initialization. Importantly, we highlight in Fig. 8 that this is also reflected in how well each approximation matches the accuracy of the predictions of the real neural network: while the small errors of the telescoping model lead to no visible differences in accuracy compared to the real neural network, using the Taylor expansion around the initialization leads to significantly different accuracy later in training.

## D.2 Additional results for case study 1: Exploring surprising generalization curves and benign overfitting

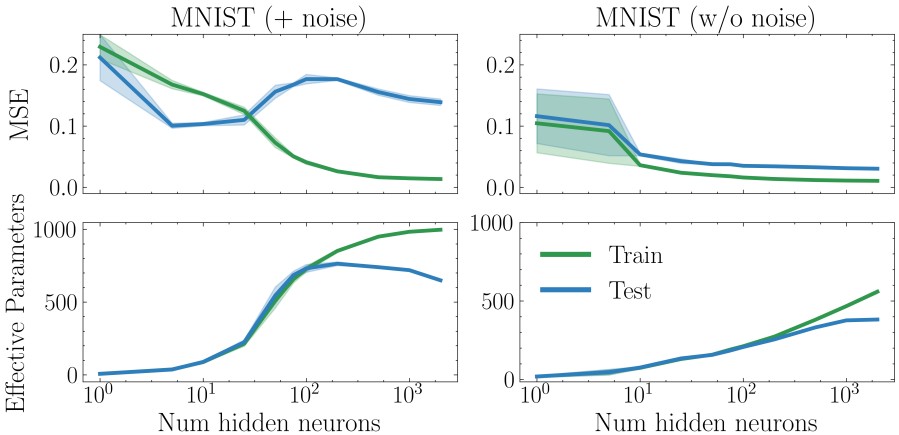

Figure 9: **Double descent** experiments using MNIST, distinguishing 3-vs-5, with 20% added label noise during training (left) and no added label noise (right). Without label noise, there is no double descent in error on this task; when label noise is added we observe the prototypical double descent shape in test error.

**Double descent on MNIST.** In Fig. 9, we replicate the CIFAR-10 experiment from the main text while training models to distinguish 3-vs-5 on MNIST. We find that in the absence of label noise, no problematic overfitting occurs for any hidden size; both train and test error monotonically improve with increased width. Only when we add label noise to the training data, do we observe the

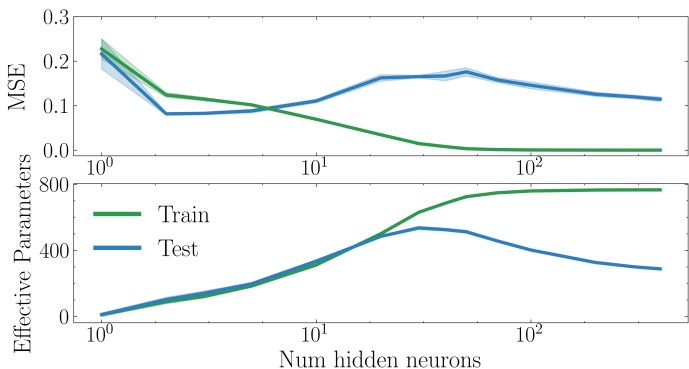

Figure 10: **Double descent** experiment using MNIST-1D, distinguishing class 0 and 1, with 15% added label noise during training. Mean squared error (top) and effective parameters (bottom) for train and test examples by number of hidden neurons.

characteristic double descent behavior in error – this is in line with [NKB+21]'s observation that double descent can be more pronounced when there is noise in the data. Importantly, we observe that as in the main text, the improvement of test error past the interpolation threshold is associated with the divergence of effective parameters used on train and test data. In Fig. 10 we additionally repeat the experiment using the MNIST-1D dataset with 15% labelnoise as in [GK24], and find that the decrease in test error after the interpolation threshold is again accompanied by a *decrease in effective parameters* as the number of raw model parameters is further *increased* in the interpolation regime.

**Additional grokking results.** In Fig. 11, we replicate the polynomial grokking results of [KBGP24] with additional values of $\epsilon$. Like [KBGP24], we observe that larger values of $\epsilon = 0.5$ lead to less delayed generalization. This is reflected in a gap between effective parameters on test and train emerging earlier. With very small $\epsilon = .05$, conversely, we even observe a double descent-like phenomenon where test error first worsens before it improves later in training. This is reflected also in the effective parameters, where $p_{\hat{\mathbf{s}}}^{test}$ first exceeds $p_{\hat{\mathbf{s}}}^{train}$ before dropping below it as benign overfitting sets in later in training. In Fig. 12, we replicate the MNIST results with additional values of $\alpha$; like [LMT22] we observe that grokking behavior is more extreme for larger $\alpha$. This is indeed also reflected in the gap between $p_{\hat{\mathbf{s}}}^{test}$ and $p_{\hat{\mathbf{s}}}^{train}$ emerging later in training.

**Additional training results on MNIST with standard initialization.** In Fig. 13, we present train and test results on MNIST with standard initialization to supplement the test results presented in the main text. Both with and without sigmoid, train and test behavior is almost identical, and learning is orders of magnitude faster than with the larger initialization. The stronger inductive biases of small initialization, and additionally using sigmoid activation, lead to much lower learned complexity on both train and test data as measured by effective parameters.

### D.3 Additional results for Case study 2: Understanding differences between gradient boosting and neural networks

In Fig. 14, we replicate the experiment from Sec. 4.2 on three further datasets from [GOV22]'s tabular benchmark. We find that the results match the trends present in Fig. 5 in the main text: the neural network is outperformed by the GBTs already at baseline, and the performance gap grows as the test dataset becomes increasingly more irregular. The growth in the gap is tracked by the behavior of the normalized maximum kernel weight norm of the neural network's kernel. Only on the `california` dataset do we observe a slightly different behavior of the neural network's kernel: unlike the other three datasets, $\frac{\frac{1}{T}\sum_{t=1}^{T}\max_{j\in\mathcal{I}_{test}^{p}}||\mathbf{k}_t(x_j)||_2}{\frac{1}{T}\sum_{t=1}^{T}\max_{i\in\mathcal{I}_{train}}||\mathbf{k}_t(\mathbf{x}_i)||_2}$ stays substantially below 1 at all $p$; this indicates that there may have been examples in the training set that are irregular in ways not captured by our experimental protocol. Nonetheless, we observe the same trend that $\frac{\frac{1}{T}\sum_{t=1}^{T}\max_{j\in\mathcal{I}_{test}^{p}}||\mathbf{k}_t(x_j)||_2}{\frac{1}{T}\sum_{t=1}^{T}\max_{i\in\mathcal{I}_{train}}||\mathbf{k}_t(\mathbf{x}_i)||_2}$ increases in relative terms as $p$ increases.

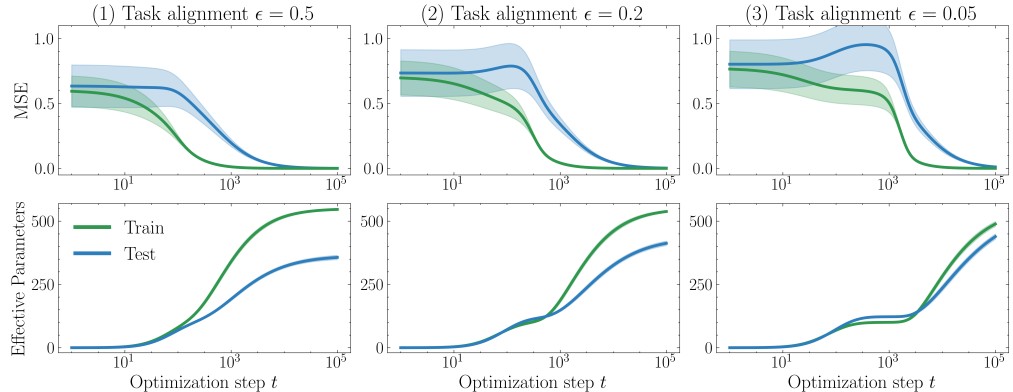

Figure 11: **Grokking** in mean squared error (top) on a polynomial regression task (replicated from [KBGP24]) against effective parameters (bottom) with different task alignment parameters $\epsilon$.

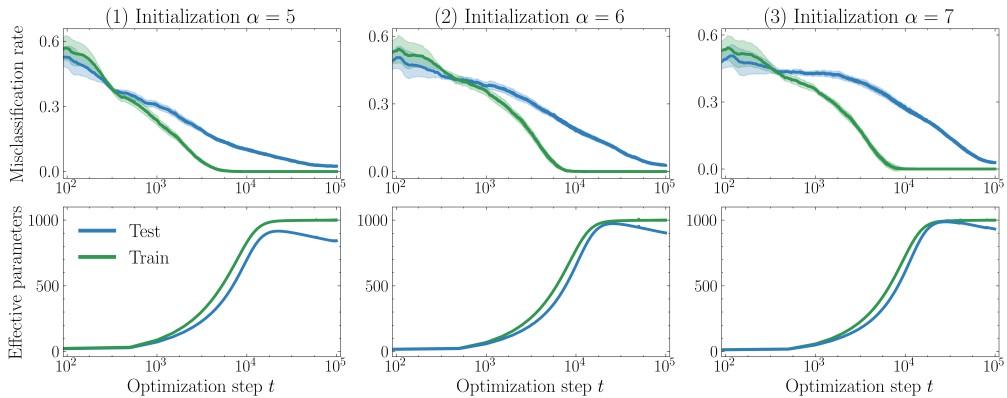

Figure 12: **Grokking** in misclassification error on MNIST using a network with large initialization ( replicated from [LMT22]) (top), against effective parameters (bottom) with different initialization scales $\alpha$.

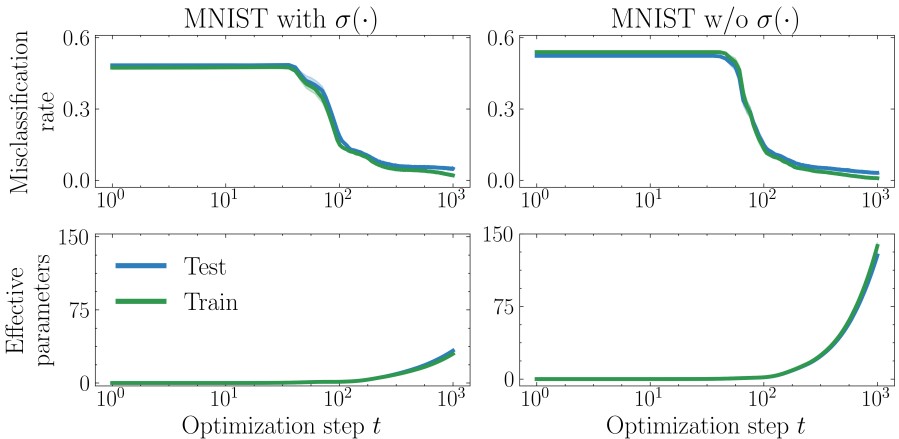

Figure 13: **No grokking** in misclassification error on MNIST (top), against effective parameters (bottom) using a network with standard initialization ($\alpha = 1$) with and without sigmoid activation.

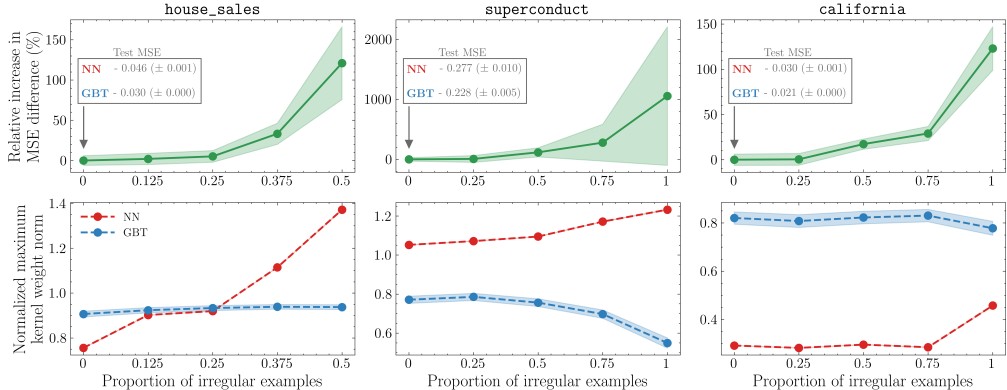

Figure 14: **Neural Networks vs GBTs:** Relative performance (top) and behavior of kernels (bottom) with increasing test data irregularity for three additional datasets.

