# OpenReview forum: "Deep Learning Through A Telescoping Lens: A Simple Model Provides Empirical Insights On Grokking, Gradient Boosting & Beyond"
_NeurIPS.cc/2024/Conference — NeurIPS 2024 poster_

### Official Review · Reviewer_EFqT · 2024-07-12

**Soundness:** 3
**Presentation:** 3
**Contribution:** 2
**Rating:** 6
**Confidence:** 2

**Summary:**

The paper proposes a new approach to understanding neural networks by examining a model consisting of a sequence of first-order approximations telescoping into an empirically operational tool. This model aims to isolate components of the training process and offers a lens to analyze the effects of design choices in neural networks. The authors illustrate the applicability of this model through case studies on double descent, linear mode connectivity etc.

**Strengths:**

The paper introduces a novel telescoping model that bridges the gap between theoretical and empirical research in neural networks.

The approach is used to derive new insights into well-known phenomena such as double descent.

The paper covers a wide range of phenomena and provides a detailed analysis of each, contributing significantly to the understanding of neural network behavior.

The model offers a simplified yet effective way to understand the impact of various design choices on neural network training.

**Weaknesses:**

The telescoping model involves increased computational costs, especially for large networks and datasets, which may limit its practical applicability.

The quality of the approximation depends heavily on the learning rate and the optimizer choice, which might not generalize well across different settings.

While the model is intended to simplify understanding, the incremental approximation can become complex to implement and analyze for certain networks and tasks.

Although the paper provides case studies, more extensive empirical validation across diverse datasets and architectures could strengthen the claims.

**Questions:**

In what ways does this model bridge the gap between purely theoretical analysis and empirical observations in neural network research?

How does the model handle non-linearities and complex interactions within neural networks? Can the authors provide a quantification of how these features affect their approximation?

Can you elaborate on how the telescoping model reveals linear mode connectivity within the loss landscape?
What are the implications of this connectivity for network training and generalization?

---

> ### Author Rebuttal · Authors · 2024-08-06
>
> We thank this reviewer for their in-depth review and their positive assessment of our work! We were delighted by their appreciation of the utility of the telescoping model itself, the wide range of phenomena we are able to consider using our model and the insights we provide. We respond to major points raised in the review below.
>
> **(1) Computational burden.** We agree that, as highlighted upfront in Sec 3, the telescoping model is more expensive to obtain than a standard neural network and therefore may not be possible to implement for large scale architectures such as modern LLMs. However, we note that the model is sufficiently tractable that it could be applied across all three case studies on the standard benchmarks from existing literature (all of which have received substantial attention in the recent literature despite their relatively moderate scale), and we were able to utilize it to provide significant insight. We believe that exploring computationally efficient methods of extending this approach to larger scale settings provides an exciting direction for future work (as alluded to in l.123)
>
> **(2) Approximation quality.** We also appreciate that, with the exception of models for which the first-order Taylor expansion holds exactly, the approximation quality will _always_ deteriorate for a sufficiently high learning rate (note that this is effectively by definition as eqn (2) holds exactly as $\gamma \to 0$). Indeed, making this potential limitation explicit was the purpose of including the experiment with varying learning rates in Fig 1.  However, we generally found the appropriate learning rates for provided experiments to achieve good approximation quality (across different optimization choices, as further investigated in Apx D.1).
>
> **(3) Increased complexity over linear models.** Indeed, we sacrifice some of the analytic simplicity of the fully linearized lazy model in order to obtain a more accurate model of a neural network. Unfortunately, we believe this to be unavoidable as a linear model will generally be unable to accurately model a full neural network in standard applied settings (indeed including even the most of toy-ish settings we consider, e.g. MNIST in Fig 1). We would argue that any conclusions drawn from a proxy model can only be valid if that proxy is a good approximation. Therefore, we believe the telescoping model sacrifices only the minimal degree of simplicity necessary in order to maintain an accurate approximation of the full network.
>
> **(4) Experiments.** Allow us to re-emphasize that the goal of this work was to provide a tool that can help us to better understand _specific previously observed_ phenomena in deep learning. Therefore, in our experimental evaluation we primarily revisit the specific, well-known, canonical experimental settings in which the studied phenomena have been observed in previous work (e.g. grokking, double descent, LMC). We have rigorously pursued this approach by reproducing key experiments from several works (e.g. [BHMM19, LMT22, KBGP24, FDP+20]) and integrating aspects of the telescoping model to better understand their findings.  Further, Apx D includes additional results for different settings of some of the case studies. We also appreciate that our concise presentation of results in the main text (which are reinforced by 4 further pages of experiments in Apx D) may have disguised the empirical contribution of this work, making it appear that our evaluations are less extensive on the surface. In the general comment, we do provide several additional results on different datasets and using different experimental settings, which were specifically requested in other reviews and further reinforce the findings of our paper. We hope that these additional results may also further reassure this reviewer.
>
> **Questions.**
> - Q1. Related to (3) above, important theoretical progress in understanding neural networks has emerged from the NTK literature. Many of these results require some assumptions e.g. infinite width limit, lazy learning, etc. The telescoping model provides a setting in which an arbitrary neural network and optimization routine can be iteratively cast using NTKs without large deviations from the exact network. We envision this can provide a bridge between theoretical and empirical research upon which future theoretical research may be able to build by applying tools from this literature to this iterative approximation instead.
> - Q2. Indeed, a powerful component of the telescoping model is that it does not require us to discard complex aspects of the optimization process or architecture. As shown in Sec 3.1, important practical extensions to vanilla SGD (i.e. momentum, weight decay, Adam) can be explicitly derived such that they can be exactly integrated into the telescoping model. Complexities of the model architecture, such as nonlinearities, on the other hand are entirely captured by the tangent features $\nabla_\theta f_{\theta_t}(x)$. That is, these gradients of the model outputs with respect to the model parameters capture how the networks architecture propagates between input and output.
> - Q3. We agree that it is an interesting next step to seek insights on the nature of loss landscapes through insights derived from LMC. In this case, we are hesitant to make strong general claims about loss landscapes on the basis of our findings. While it is clear from eqn (10) that constant gradients are a _sufficient_ condition for LMC, in practice we cannot broadly claim that gradients will become _exactly_ constant and a complete theory of LMC may need to take into account further intricacies of neural networks. We hope that a more complete theory of LMC can be developed through future research taking into account the telescoping insights we provide here, which would provide valuable insights into the nature of loss landscapes in deep learning.

---

### Official Review · Reviewer_gQrQ · 2024-07-17

**Soundness:** 1
**Presentation:** 1
**Contribution:** 1
**Rating:** 3
**Confidence:** 3

**Summary:**

This paper tries to understand the design of optimizer, model architectures and some deep learning phenomenon empirically. Additionally, the authors find a proxy model of neural networks other than the simply linearized model.

**Strengths:**

1. The topic is interesting, especially the proxy model of neural networks. I would like to see more theoretical insights derived from the proxy model, but the current paper didn't put efforts on it.

**Weaknesses:**

1. First of all, limited insights gained from the proxy model of neural networks when dealing with the modern optimizer and architectures. In section 3.1, the authors try to connect their proposed proxy model with some modern optimization techniques. However, no rigorous proof or insightful experiments are provided. Complex formulas including Eq. 6-8 only provides limited insights.
2. As for case study part, the writing confuses me a lot. For example, in Section 4.1, in line 200-201, "A candidate complexity measure that avoids the shortcomings listed above because it only considers the behavior of the final fitted model was recently used by [CJvdS23] in their study of non-deep double
descent." I cannot understand what this mean.
3. Still the case study part, I strongly disagree that "LMC and train-time transition into lazy regime." (As I am not familiar with GBT and grokking, I skimmed those parts). [cite 1] and [cite 2] demonstrates that lazy regime is not sufficient to explain LMC and model merging.

[cite 1] Fort, Stanislav, Gintare Karolina Dziugaite, Mansheej Paul, Sepideh Kharaghani, Daniel M. Roy, and Surya Ganguli. "Deep learning versus kernel learning: an empirical study of loss landscape geometry and the time evolution of the neural tangent kernel." Advances in Neural Information Processing Systems 33 (2020): 5850-5861.
[cite 2] Ortiz-Jimenez, Guillermo, Alessandro Favero, and Pascal Frossard. "Task arithmetic in the tangent space: Improved editing of pre-trained models." Advances in Neural Information Processing Systems 36 (2024).

4. There are seeming no many experiments in both case study part and Sec 3. I would expect more experiments relating their proposed proxy model to the case studies and optimization techniques.

**Questions:**

N/A

---

> ### Author Rebuttal · Authors · 2024-08-06
>
> We thank the reviewer for the feedback. We are delighted that the reviewer finds our proxy model for a neural network interesting! Before we move to address the specific points raised in the review below, we would like to make two high-level comments.  First, while we similarly believe that interesting theoretical insights can be derived from this model in follow-up work, we would like to note that the _explicitly stated goal_ of the current paper was to use it as a tool for generating new _empirical_ insight, unifying the study of a range of modern deep learning phenomena. Second, we also find it unfortunate that Secs 4.1 and 4.2 were only "skimmed" and therefore barely feature in the review, and would like to highlight that these were the parts that received particular praise in all other reviews.
>
> **1. Insights from Sec 3.1.** First, we note that as stated in L131, eqs 6-8 were not only included to provide insight, but were also _necessary_ in order to replicate later experiments from existing work that required Adam, weight decay, and momentum to be applied in the telescoping model. Further, we believe that Sec 3.1 will be of interest to anyone wishing to gain deeper _intuition_ into the effect different optimization strategies will have _on model predictions_. Usually, these strategies are only discussed in terms of their effect in parameter space, and not in function space as we do here – even though function space is what ML is ultimately interested in. While we appreciate that this reviewer was not interested in the insights derived from Sec 3.1, we note that both YCvv and b22S explicitly mentioned this section as a strength of the paper. We hope that in examining the new empirical results we have now provided (see the pdf in the top-level response) that demonstrates how these insights are reflected in training a real neural network this reviewer will also find value in Sec 3.1.
>
> **2. Clarifying cited paragraph on complexity.** This sentence highlights that the cited complexity metric of [CJvdS23] accounts for what the model has learned during the fitting process, and is thus able to capture the interaction of model, optimizer and a particular dataset. Many other metrics only consider what is _learnable_ by the entire hypothesis class for arbitrary data. We note in the opening paragraph of this section that the former type of metric is preferable for understanding double descent and grokking, where the phenomena only arise due to interaction of the model, its learning process and the data. We hope that this alleviates any confusion. We would also be happy to clarify any further unclarities in the writing if more precise issues were pointed out!
>
> **3. LMC.** We thank this reviewer for raising this question and are happy to clarify why there is no contradiction between these works and ours.
> - Fort et al., which we cite numerous times as [FDP+20], find that “deep network training exhibits a highly chaotic rapid initial transient” for the first 2-3 epochs after which the relative change in the NTK decreases throughout training and plateaus at a low value. This is in complete agreement with our work which studies this through the lens of the telescoping model. Note that our Fig 5 and their Fig 7 - which are similar in spirit but study slightly different objects and use different metrics - observe a very similar trend. We further supplement their results by examining differences in tangent feature changes between pretrained and randomly initialized models.
> - Ortiz-Jimenez et al. focus on a very different setting: they consider the context of task arithmetic, where a generalist model is being finetuned several times, each time on a new dataset, where the input space $\mathcal{X}$ of each task is disjoint from the others, with resulting task vectors then _added_ to the base models weights and, therefore, is subtly different to _weight averaging_ (they consider addition and subtraction). Please note that we strictly focus on finetuning routines that only differ in “batch orderings and data augmentations” (L356) followed by weight averaging between the resulting models, and do not claim that our empirical findings will transfer to their setting. Nonetheless, we agree that this reference is relevant to include and will discuss it in the updated version! Thank you for pointing this out.
>
> **4. Experiments.** The goal of this work was to provide a tool that can help us to better understand _specific previously observed phenomena in deep learning_. Therefore, in our experimental evaluation we primarily revisit the exact well-known, canonical experimental settings in which the studied phenomena have been observed in previous work (e.g. grokking, double descent, LMC). We have rigorously pursued this approach by reproducing key experiments from several works (e.g. [BHMM19, LMT22, KBGP24, FDP+20]) and integrating aspects of the telescoping model to better understand their findings. We also verified that the telescoping approximation holds for the different implementation choices discussed in Sec 3.1 in Apx D.1.
>
> We also appreciate that our concise presentation of results in the main text (which are reinforced by 4 further pages of experiments in the Appendix) may have disguised the empirical contribution of this work, making it appear that our evaluations are less extensive on the surface. For example, the double descent experiments alone - which only constituted a part (Fig 2) of case study 2 - required training 88 models per setting resulting in 432 hours of compute time on an A100 GPU.
>
> Finally, in the pdf in the general comment we provide several additional results which were specifically requested in other reviews and further reinforce the findings of our paper. In particular, we also provide some empirical verification of our expressions from Sec 3.1 which we hope will further reassure this reviewer.

---

### Official Review · Reviewer_b22S · 2024-07-18

**Soundness:** 3
**Presentation:** 2
**Contribution:** 3
**Rating:** 6
**Confidence:** 3

**Summary:**

The paper replaces the traditional "lazy learning" regime approximation with an approximation using a telescoping sum. This splits up the usual interval of approximation $\theta_T - \theta_0$ into many smaller pieces. The authors then show why this approximation is useful for explaining various interesting aspects of modern deep learning:
1. Generalisation behaviour (double descent, grokking and the effect of inductive bias on model complexity)
2. The connection between gradient boosting and modern deep neural networks (DNNs)
3. The success of weight averaging
4. Modern design choices regarding architecture and optimiser

**Strengths:**

In general, I think the authors did a reasonably good job. Particular praise goes to the following aspects of the work:
1. The extension of the lazy regime approximation to a telescoping sum seems like a very natural extension of the current set of ideas. The use of the sum seems logical and may become a useful theoretical insight.
2. I liked the choice of case studies. Phenomena like grokking, double descent, and weight averaging are clearly of interest to the community.
3. The explanations provided seem relatively intuitive under the telescoping approximation.

I will now touch on how well I think the paper does on the areas of originality, quality, clarity, and significance:
1. *Originality.* The use of a telescoping sum is clearly somewhat original. In addition, I thought the analysis of GBTs seemed fairly novel and interesting.
2. *Quality.* There are some elements of high-quality work in the paper. I think the analysis and empirical evidence for the generalisation section were fairly good. I also thought the analysis of momentum and weight decay seemed interesting. (However, I have issues/suggestions for both of these sections in weaknesses).
3. *Clarity.* I thought the authors did a fairly good job of justifying their use of a telescoping sum. However, I find it hard to come up with other strengths, as I think clarity is what this paper is most lacking.
4. *Significance.* I think the insights are of decent significance. For example, if it was demonstrated that the effective parameter count metric had further predictive power, it could be a useful tool in the analysis of grokking. (I provide suggestions as to how it could be expanded upon).

**Weaknesses:**

I will structure this section into three subsections. The first two subsections concern general where I believe the paper may be lacking. In the third section, I list additional minor issues. I then end with a general assessment.

### (1) Lack of clarity and writing style

I think the paper lacks clarity in the writing. Here are some major areas where this causes issues:
1. **Abstract.** The abstract does not describe the idea of the paper well nor provides a clear explanation of the impact of the work. I found it very hard to read at first.
2. **Explanation of insights.** Due to the density of the writing, I found it hard to understand the theoretical insights.
3. **Explanation of setup in section 4.1.** I found the explanation of the $p_{\hat{s}}^0$ measure to be quite confusing. I could not see where $\hat{s}(x)$ was clearly defined.

### (2) Importance of theoretical insights and empirical backing

In section 3.1, there are several interesting observations about modern optimisation choices and how they can be more easily reconciled under the telescoping approximation. However, I am unsure of several things:
1. **Utility of telescoping sum.** Are the observations provided specific to the telescoping sum approximation or would they hold with the traditional lazy approximation?
2. **Claims in real networks.** Do the claims made hold in real networks? It would have been nice to see experiments demonstrating the claims made given that we are still working with an approximation.
3. **Utility of insights.** What should I do with the insights provided (supposing they hold in real cases)? For example, now that I know that momentum offsets the effect of weight decay on learned function updates, should I try and avoid using weight decay and momentum? In my experience, it seems that weight decay and momentum do work pretty well together in practice.

In section 4, there are also many interesting observations. However, I have a few concerns:
1. I would have liked to have seen more experiments demonstrating that cases of double descent and effective parameter counts are related in the way described. I think it is likely to be true, I just don't think that one experiment is sufficient.
2. In the grokking cases, I would like to see whether the metric proposed (namely mismatch in the effective parameter count) has more predictive power. For example, in the literature, there have been ways of inducing grokking proposed. For example, in [this paper](https://openreview.net/pdf?id=ux9BrxPCl8) it is shown that adding spurious dimensions to the input reliably increases grokking. I would like to see whether cases with a greater grokking gap correlate with a greater divergence in effective parameter use.
3. In the second case study, the theory of why GBTs perform better than neural networks seems reasonable. However, I think the empirical evidence is lacking. A single plot with correlation does not seem sufficient.
4. In the third case study, I am unsure as to whether the trend in the pre-activation gradients aligning with the accuracy gap is sufficient to demonstrate the hypothesis.

### (3) Minor gripes with the paper

1. If I have understood correctly, the paragraph discussing approximation error seems unnecessary. I would place this in an appendix and spend two sentences on it in the main discussion. It seems quite obvious that the telescoping sum is going to approximate things better.
2. The practical limitations section also seems unnecessary. We know that it is a tool for analysis rather than an alternative approach for training. This does not need to be explained.
3. I'm not sure how I feel about the use of the word "pedagogical" throughout the paper. I don't know if the impact of a NIPS paper is meant to be a tool for teaching. Rather, the desired impact should be a novel result or an improvement to the understanding of the field.

### A general assessment

Generally, I think the paper provides some interesting insights but is attempting to do too much. I think if it were to be rejected, I would encourage the authors to rewrite it in a much simpler style focused on far fewer topics with many more experiments.

Update: I have moved my overall rating from a 5 to a 6 as I believe some of my concerns were addressed in the rebuttal. I also moved my rating of soundness from a 2 to a 3.

**Questions:**

Some questions:
1. As someone who has read relatively few papers using tangent kernel arguments, I am curious how other groups have extended the traditional lazy -> rich approximation?
2. Could the authors provide a better explanation of $p_{\hat{s}}^0$ and associated nomenclature? What is $\mathcal{I}_0$ for example?
3. Are there meaningful experiments to run with the insights in section 3.1? For example, could the authors look at verifying the reported trade-off between momentum and weight decay?

Some other suggestions:
1. A few grokking papers which are not included in the literature review are:
   - [Paper 1](https://openreview.net/forum?id=GH2LYb9XV0). Grokking in linear estimators without regularisation, which may go against intuitions with the measure proposed
   - [Paper 2](https://openreview.net/forum?id=ux9BrxPCl8). Discusses complexity as a theory of grokking.
   - [Paper 3](https://openreview.net/forum?id=UHjE5v5MB7). Talks about corrupted labels, might be interesting in discussion of future work where the proposed measure could be used to analyse cases with corrupted labels
2. Paper on double descent not mentioned: [Double Descent Paper](https://arxiv.org/pdf/2111.08234).

**Limitations:**

I think limitations are appropriately addressed given that the paper is not focused on performance. It might be interesting to see cases where the telescoping sum does not approximate some set of phenomena well. Perhaps some discussion of this instead of the current limitations paragraph would be more meaningful.

---

> ### Author Rebuttal · Authors · 2024-08-06
>
> Thank you for the very detailed and constructive review! Limited by space constraints, we respond to major comments below.
>
> **(1) Clarity.** We appreciate that our writing was somewhat densely packed in the mentioned sections, and will utilize the camera-ready extra page to decompress for improved clarity. We will particularly expand the paragraph on $p^0_s$ and will more explicitly write out $\hat{s}(x)$ in the main text (currently derived for different optimizers in Apx B.1), which can recursively be obtained from eq 5 after substituting in the squared loss gradient. We will also rework the abstract to include a more explicit summary of the finding of each case-study.
>
> **(2) Comments on Sec 3.1.**
> 1. The insights on functional updates that can be derived from the telescoping sum subsume those of the lazy regime as a special case. As we highlight in the paragraphs on momentum and weight decay, assuming constant gradients will simplify some of these expressions. Some observations could thus also be made when assuming lazy learning – e.g. adaptive learning rates acting by rescaling the kernel – however, we have not seen these observations discussed explicitly anywhere. We will make even more clear which observations also hold in the lazy regime!
> 2. _Claims in real networks._ This is point is precisely why we included experiments on approximation quality – Fig 1 and Fig 6 in Apx D show that the approximation matches real networks very closely for the different implementation choices and that the insights in Sec 3.1 thus apply to real networks. We appreciate the suggestion to additionally numerically verify the tradeoff between momentum and weight-decay empirically, and have included an experiment that does so in the pdf in the top-level response!
> 3. _Utility of insights._ We believe that Sec 3.1 will be of interest to anyone wishing to gain more intuition and understanding of the effect different optimization strategies will have on model predictions. Usually, these are only discussed in terms of their effect in parameter space, and not in function space as we do here – even though function space is what ML is ultimately interested in! We comment further on the specific momentum weight decay example in the experiment in the top-level response.
>
> **(3) Comments on Sec 4.**
> 1. _Double descent._ Note that Apx. D.2 already includes additional experiments, indeed demonstrating the same behavior for double descent on MNIST. Additionally, we provided a new experiment in the pdf of the top-level response.
> 2. _Grokking._ In Apx. D.2, we actually already conducted additional experiments varying two such mechanisms –  the initialization scale [LMT22] and task alignment [KBG24]. As discussed in Apx D.2  we found in both cases that later generalisation (i.e. more extreme grokking) is predicted by an effective parameter gap emerging later in training. As suggested in the review, we repeated this experiment for another mechanism – the number of spurious dimensions – on the parity prediction task of Miller et al (TMLR2024). The results, included in the top-level comment, indicate that also here grokking is predicted by the effective parameter gap growing. We will better link to this in the updated main text, thank you for the suggestion!
> 3. _GBTs._ Note that results using additional datasets are included in Apx D.3 where a similar behavior is observed in the kernels.
> 4. _LMC._ We agree that unlike the preceding sections, the case study in Sec 4.3 is more exploratory in nature. The key point of interest is using the telescoping lens to come to the observation in eq 10 that constant gradients are sufficient for LMC. However, in practice gradients will not reach perfect stabilization and therefore this cannot provide a complete explanation for this phenomenon. We were therefore deliberate in not claiming any overly strong conclusions, and only include preliminary evidence of a specific factor contributing to LMC emerging by empirically showing that the model gradients indeed become closer to stable throughout training, and that – like the loss barrier – the change in model gradients is lower in pretrained than in randomly initialized models. We think there is significant opportunity to further investigate this perspective, and will update the section to be more explicit on scope and potential future avenues.
>
> **(4) Minor points.**
>
> 1. _Approximation quality section (Fig 1)._ Building on the answer to point 2 on Sec 3.1 above, we found this section essential as it demonstrates that (and when) the telescoping sum is indeed a reasonable model of a neural network in a practical setting [note that the neural networks used here are the same as in Sec 4.1] and that therefore the derived insights hold, while those of the lazy model may not. Further, the main purpose of including the lazy model here was to provide a _reference point_ for the scale of the approximation error, allowing us to better evaluate how good an approximation the telescoping model is. We will make the purpose of this section more clear in the updated manuscript.
> 2. _Limitations._ We found it important to be upfront about potential computational limitations. The comment on training is in reference to the lazy model, which _is_ sometimes used as an alternative way of training (as a linear regression in the tangent features). We will make this more clear!
> 3. _The term pedagogical._ We are in full agreement that the goal of a NeurIPS paper should be an ``improvement to the understanding of the field''. Our intention was to imply the telescoping model to be a pedagogical device in the sense that it is a tool through which researchers can better understand aspects of deep learning. However, we are happy to substitute this term as this was obviously not clear.
> 4. _Additional relevant references._ Thank you for these, we will include them in our literature review!

---

> > ### Comment · Reviewer_b22S · 2024-08-07
> > **Response to rebuttal for b22S**
> >
> > I would like to thank the authors for their detailed response, it cleared up many of my questions and concerns. As a result, I will update my rating to 6 (weak accept) and my assessment of soundness to 3.
> >
> > A few lingering questions and comments:
> > 1. Could the authors explain in a comment what is meant by $\hat{s}(x)$, unfortunately, I was not able to understand its meaning from Appendix B.2.
> > 2. I still believe the section on the approximation performance is not really necessary. My original concern was that even if the approximation holds with reference to the way the learning trajectories look, the approximation might not hold when used to derive other properties with the telescoping sum. (I hope this makes sense!)

---

> > > ### Author Response · Authors · 2024-08-08
> > > **Response [1/2]**
> > >
> > > We thank the reviewer for the very quick response and for engaging with our rebuttal – we really appreciate it! We are delighted that our rebuttal alleviated most concerns and hope that our responses below address any leftover reservations.
> > >
> > > 1. **What is  $\mathbf{s}\_{{\theta}\_{T}}(x)$?**
> > >
> > > _Motivation._  [CJvdS23] showed that their _effective parameter_ metric $p^0\_s$ can be used to explain double descent in some _non-deep_ models (linear regression and tree-based methods) because it allows to distinguish between train-time and test-time complexity. In section 4.1, we explore whether the telescoping model allows us to make use of $p^0\_s$ in deep models too, which is not obvious a priori as the use of $p^{0}\_s$ requires to be able to write model predictions in smoother form.
> > >
> > > *Extended background.* [CJvdS23] build on ideas from the literature on nonparametric regression smoothers, which are methods that output predictions that are a _linear combination_ of the nx1 vector of training labels $\mathbf{y}$. That is, such methods issue predictions $\hat{f}(x)=\hat{s}(x)\mathbf{y}$ where $\hat{s}(x)$ is a 1xn vector assigning a weight to each training example. Prototypical examples of (the very broad class of) smoothers are k-Nearest Neighbor (kNN) estimators, which use smoother weights $\hat{s}(x) \in \\{0, 1/k\\}^n$, and ordinary linear regression (OLS), which uses smoother weights $\hat{s}(x)=x’(X’X)^{-1}X’$. Effective parameters $p^{0}\_{s}=\frac{n}{|\mathcal{I}\_0|}\sum\_{j\in \mathcal{I}\_0} ||\hat{s}(x\_j)||^2$ then aim to make different types of smoothers comparable to each other w.r.t. the amount of smoothing they do, by measuring the average squared norm of the smoother weights when issuing predictions for a given set of inputs $\\{x_j\\}\_{j \in \mathcal{I}\_0}$. Here $\mathcal{I}\_0$ is simply used as a shorthand to collect the indices of the inputs for which the used effective parameters are measured; e.g. $\mathcal{I}\_{train}=\\{1, \ldots, n\\}$ the indices of the training data and  $\mathcal{I}\_{test}=\\{n+1, \ldots, n+m\\}$ the indices of the test data. Intuitively, $p^0\_s$ provides a measure for how non-uniform and extreme the learned smoother weights are – the higher $p^{0}\_{s}$, the more complex the model. For example, OLS with $p$ covariates has $p^{0}\_{s}=p$ and a kNN estimator has $p^{0}\_{s}=\frac{n}{k}$.
> > >
> > > *How does the telescoping model enable us to use $p^0\_s$?* For the squared loss, we exploit that the SGD functional updates $\Delta \tilde{f}\_t(x) = \gamma \nabla\_{{\theta}} f\_{{\theta}\_{t-1}}(x)^\top \mathbf{T}\_t(\mathbf{y} - \tilde{\mathbf{f}}\_{\theta\_{t-1}})$ use only a linear combination of the training labels, which is how we recursively construct $\mathbf{s}\_{\theta\_t}(x)$. To see this, assume for simplicity that at initialization $f\_{\theta\_0}({x})=0$ (otherwise, the prediction at initialization is carried forward as additional constant term $c^0\_{\theta\_t}(x)$ in the paper). Then, the first functional update is $$\Delta \tilde{f}\_1(x)=\gamma \nabla\_{\theta} f\_{\theta_{0}}(x)^\top \mathbf{T}\_1(\mathbf{y} - \mathbf{0}) =\underbrace{\gamma \nabla\_{\theta} f\_{\theta\_{0}}(x)^\top \mathbf{T}\_1}\mathbf{y}  = \mathbf{s}\_{\theta\_1}(x)\mathbf{y}$$ which means that $\tilde{f}\_{\theta\_1}(x)=0 +\mathbf{s}\_{\theta\_1}(x)\mathbf{y}$ is also a linear combination of the training labels. Letting $\mathbf{S}\_{\theta\_1}$ denote the nxn matrix that has as the ith row $\mathbf{s}\_{\theta\_1}(x_i)=\gamma \nabla\_{\theta} f\_{\theta\_{0}}(x_i)^\top \mathbf{T}\_1$ for the ith training example and $\mathbf{I}\_n$ denote the nxn identity matrix, we note that the second update can then be written as $$\Delta \tilde{f}\_2(x) = \gamma \nabla\_{\theta} f\_{\theta\_1}(x)^\top \mathbf{T}\_2(\mathbf{y} - \mathbf{S}\_{\theta\_1}\mathbf{y}) = \underbrace{\gamma \nabla\_{\theta} f\_{\theta\_1}(x)^\top \mathbf{T}\_2(\mathbf{I}\_n - \mathbf{S}\_{\theta\_1})}\mathbf{y}= \Delta \mathbf{s}\_{\theta\_2}(x)\mathbf{y}$$ which is also a linear combination of the training labels. Then, by adding the functional update to the prediction from the first step, we get the prediction at the second step $$\tilde{f}\_{\theta_2}(x)=0 +\mathbf{s}\_{\theta\_1}(x)\mathbf{y} +\Delta \mathbf{s}\_{\theta\_2}(x)\mathbf{y} = (\mathbf{s}\_{\theta\_1}(x) +\Delta \mathbf{s}\_{\theta\_2}(x))\mathbf{y} = \mathbf{s}\_{\theta\_2}(x) \mathbf{y}$$ which is again a linear combination of the training labels. Then, by recursion, any future update analogously adds a term $$\Delta \mathbf{s}\_{\theta\_t}(x) = \gamma \nabla\_{\theta} f\_{\theta\_{t-1}}(x)^\top \mathbf{T}\_t(\mathbf{I}\_n - \mathbf{S}\_{\theta\_{t-1}})$$ to the previous smoother weight. Finally, this then gives $$\mathbf{s}\_{\theta\_T}(x)=\sum^T\_{t=1}\Delta \mathbf{s}\_{\theta\_t}(x)$$ where the $\Delta \mathbf{s}\_{\theta\_t}(x)$ depend only on the gradient feature matrices $\\{\mathbf{T}\_{t'}\\}_{t'\leq t}$.
> > >
> > > [Continued in next comment below]

---

> ### Author Response · Authors · 2024-08-08
> **Response [2/2]**
>
> [Continued from above]
>
>  As derivation of  $\mathbf{s}\_{\theta\_T}(x)$ requires only straightforward yet somewhat tedious algebra, we had relegated it to the Appendix to ensure readability of Section 4.1. However, we now see that the reader could benefit from some of the above discussion in the main text to make it more self-contained. As indicated in our rebuttal, we are very grateful for the suggestion and will use some of the additional space in the camera-ready to expand the corresponding paragraph on page 5.
>
> 2. **Approximation performance.**
>
> We recognize the first point being made by the reviewer - the telescoping model is strictly a better approximation of a neural network than the lazy model _by design_ as the difference is simply taking smaller approximation terms. This point does not require major discussion in the main text. The more important aspect of this section is demonstrating that, at least on the tasks evaluated, the difference in approximation quality results in a categorical improvement from a model that certainly does not track the performance of a neural network to one that does. The point is cemented further in Fig 7 where we observe the lazy model catastrophically diverging from the neural network’s prediction performance early in training while the telescoping model matches performance throughout. We would argue that this categorical difference in the two models ability to emulate a neural network is vital to demonstrate as it provides the essential justification for accepting the added complexity of the telescoping over the lazy model (e.g. one could conceivably compute effective parameters in case study 1 using the lazy approximation instead (as it is a special case of the telescoping approximation), but these results demonstrate that it would be a mistake to expect conclusions drawn in that setting to transfer to a fully trained neural network).
>
>
> We appreciate this reviewer also has a subtle point that matching performance is _necessary_ but not _sufficient_ to know that a model is generally reliable as a proxy for analysis. One could imagine designing a model that matches a neural network relatively well in terms of predictions but that are achieved by a very different mechanism (e.g. using AdaBoost with decision trees). In that case, findings about the inner workings of the proxy model could plausibly _not_ also hold for the neural network it models. In some sense, this is an unavoidable fact of modelling any process (particularly relevant in the social sciences where natural data-generating processes are often modelled using generalized linear models). However, we would point to two strong pieces of evidence that suggest the telescoping model only induces minimal bias in its approximation. (1) Unlike in e.g. the social sciences, we _do_ have access to the exact mathematical form of the natural phenomenon (a neural network being trained on data) and are taking a direct approximation of that neural network. (2) The empirical predictions derived from the telescoping model have been reflected in the behavior of the full neural network repeatedly throughout our empirical investigations (e.g. (a) the divergence of full neural networks and gradient boosting is well modelled by the kernel of the telescoping model or (b) the predicted counterbalancing effect of weight decay and momentum from the telescoping model was demonstrated to exist exactly in a full neural network in our new results in the pdf).
>
>
> Once more, we would like to thank this reviewer for their detailed engagement with our submission. We will integrate aspects of this extended discussion into the main text as we believe they are indeed valuable to expand upon. We hope that we have provided sufficient clarification on these final concerns but would be happy to discuss further if not!

---

### Official Review · Reviewer_YCvv · 2024-07-28

**Soundness:** 3
**Presentation:** 2
**Contribution:** 3
**Rating:** 7
**Confidence:** 3

**Summary:**

This work proposes a new analytical model of neural networks (NNs) extending popular 'linearized' approximations over the initial parameters built from the gradient vectors. In particular, they consider approximating the full learning trajectory of NNs as a sequence of first-order approximations rather than a single one. The authors connect their model to several prominent empirical observations of the properties of neural networks, showing how it can be used to compare different optimizers and learning frameworks, and also justify phenomena such as grokking, double-descent, and linear-mode connectivity.

**Strengths:**

1) The authors provide a comprehensive first look into the properties and implications of the proposed framework. For instance, they consider extensions to vanilla SGD updates, including momentum and weight decay - showing how they can connect quite naturally through a weight averaging perspective (start of Sec 3.1).
2) The Appendix is functionally used to relegate more comprehensive secondary details, allowing the authors to keep the text free of long proofs and extended literature reviews. This makes the main text concise and pleasant to read, while still allowing the interested readers to delve deeper into specific topics.
3) The paper does a good job at highlighting the relevance of the proposed framework, providing grounded examples that connect it with several popular understandings of neural network dynamics and behavior in Section 4. Out of the three examples, I found the discussion regarding comparing neural networks and gradient boosting through their relative approximated dynamics and kernels (4.2) of particular interest.

**Weaknesses:**

1) The proposed telescoping approximation partially defeats the purpose of the simplified linear models, by introducing back much complexity that linearization precisely seeks to abstract away. Perhaps, some additional discussion about the limitations in terms of theoretical analysis would have been useful.
2) Also due to this increased complexity, most experiments are carried out in relatively toy-ish settings (e.g., MNIST).
3) There is some overuse of inline math notation (e.g., Section 4.1) hindering readability. While I understand the limitations due to the imposed page constraints, I would encourage the authors to try and adjust their text to lay out their equations more clearly in later revisions.

Typos:

"due rescaling" -> due to rescaling (117)

**Questions:**

I would appreciate if the authors could address the criticism highlighted above in their rebuttal.

**Limitations:**

The authors explicitly discuss the main limitation of the proposed methodology (increased complexity) at the end of Section 3, although mostly from a practical (and not theoretical) viewpoint. Overall, I found the work well-written, quite comprehensive, and insightful - building directly on top prior work, yet, providing some interesting new perspectives. Hence, I am leaning toward acceptance.

---

> ### Author Rebuttal · Authors · 2024-08-06
>
> We thank this reviewer for their very constructive review and the positive assessment of our work! We were especially delighted to read that the reviewer found our uncovered connection between the telescoping model and gradient boosting to be particularly instructive. We respond to the comments raised in the review below.
>
> **(1) Increased complexity over linear models.** Indeed, we sacrifice some of the analytic simplicity of the fully linearized lazy model in order to obtain a more accurate model of a neural network. Unfortunately, we believe this to be unavoidable as a linear model will typically be unable to accurately model a full neural network in standard applied settings (indeed including even the most of toy-ish settings we consider, e.g. MNIST in Fig 1). We would argue that any conclusions drawn from a proxy model can only be valid if that proxy is a good approximation. Therefore, we believe the telescoping model sacrifices only the minimal degree of simplicity necessary in order to maintain an accurate approximation of the full network. We appreciate the reviewer's suggestion to extend the limitations section beyond practical considerations by highlighting that this additional complexity can come at the cost of complicating potential future theoretical analyses, and will do so in the updated manuscript!
>
> **(2) Scale of experimental settings.** While we agree that our experiments were generally conducted in relatively moderate-scale settings compared to e.g. much of the recent LLM focused empirical work, we would like to emphasize that we were able to operate at a sufficiently large scale to satisfy the three provided case studies, all of which have received substantial attention in the recent literature (despite their relatively moderate scale). For case study 2 we operate on typical size tabular datasets from a standard benchmark from the literature.  For case studies 1 & 3, we primarily revisit the exact well-known, canonical experimental settings in which the studied phenomena have been observed in previous work (e.g. grokking, double descent, LMC from [BHMM19, LMT22, KBGP24, FDP+20]). While larger scale experiments were thus not necessary in order to address our selected research questions here, we certainly agree that considering how to apply the proposed telescoping model to larger scale networks would enable many interesting research directions in future work (as we allude to in l.123).
>
> **(3) Use of in-line maths.** We were delighted to read that this reviewer appreciated our efforts to make the reading experience as natural as possible by relegating extended details to the appendix for interested readers. We also appreciate that the quantity of content and breadth of phenomena covered in this work required certain parts of the paper to be written quite densely. This was a fair point raised by several reviewers. We will therefore put considerable effort into using the extra page of a camera-ready version to decompress the text (e.g. less in-line maths particularly in Sec 4.1) and expand on details where there is an opportunity to improve clarity.
>
> _Typo:_ Thank you, fixed!

---

> > ### Comment · Reviewer_YCvv · 2024-08-13
> > **Thanks for your rebuttal**
> >
> > I would like to thank the authors for their concise rebuttal which managed to address my main concerns, and for providing additional experiments and analysis. I also read the other reviews, and while I do understand some of their shared concerns e.g., "limited insights [...] when dealing with the modern optimizer and architecture" (gQrQ) I believe these have been mostly addressed/should be addressable (e.g., by toning down some of the claims), and do not warrant outright rejection. I believe expecting each paper to provide definite evidence for some of the key questions in the field is unrealistic. However, I believe this paper makes a solid effort to provide insightful novel evidence and analysis, which is likely to be of interest to the wider research community. Thus, I stand by my original assessment and would argue this paper clearly meets the bar for acceptance.

---

### Author Rebuttal · Authors · 2024-08-06

We would like to thank all reviewers for the time and effort put into the review process! We are grateful for the constructive nature of the reviews and were delighted by the largely positive assessment of our work. We were especially excited by the recurring appreciation of the new insights on (i) the effects of optimization strategies, (ii) the phenomena relating to model complexity (double descent and grokking), and (iii) the relationship between neural networks and gradient boosting, all of which the construction of the telescoping model allowed us to provide in this work.

We respond to all reviewers individually below, but re-address some recurring comments globally here. We also include additional experimental evidence in the pdf attached to this comment. Note that due to the strict 6000 character limit in rebuttals our responses were forced to be brief. However, we would be happy to expand our discussion on any point if further details are helpful!

**1. Density of writing.** We appreciate that the quantity of content and breadth of phenomena covered in this work required certain parts of the paper to be written quite densely. We will therefore put considerable effort into using the extra page of a camera-ready version to decompress the text (e.g. using less in-line maths particularly in Sec 4.1) and expand on details where reviewers pointed out that there is an opportunity to improve clarity!

**2. Further experiments.** Allow us to re-emphasize that the goal of this work was to provide a tool that can help us to better understand _specific, previously observed_ phenomena in deep learning. Therefore, in our experimental evaluation we primarily revisit the exact well-known, canonical experimental settings in which the studied phenomena have been observed in previous work (e.g. grokking, double descent, LMC). We have rigorously pursued this approach by reproducing key experiments from several works (e.g. [BHMM19, LMT22, KBGP24, FDP+20]) and integrating aspects of the telescoping model to better understand their findings.

Where specific additional experiments were requested we have provided them in the attached pdf. In particular, the pdf includes:
- Empirical verification of some of the theoretical predictions made in Sec 3.1 on the opposing effects of weight decay and momentum on model predictions using real networks obtained through standard training.
- Additional experiments on grokking in a parity prediction task with varying number of spurious features, further demonstrating that the behaviour of effective parameters indeed predicts grokking behaviour.
- A double descent experiment on a third dataset, the MNIST-1D dataset (Greydanus  \& Kobak, (ICML2024). ``Scaling Down Deep Learning with MNIST-1D.'') which was proposed recently as a sandbox for investigating empirical deep learning phenomena and was also used to demonstrate deep double descent in the textbook [Pri23].

---

### Decision · Program_Chairs · 2024-09-25

**Decision:**

Accept (poster)

**Comment:**

This paper introduces an analytics tool that consist of a sequence of first-order approximations with a telescoping sum to aid the understanding of various deep learning phenomena, as well as the explanation of several empirical observations.

Most reviewers think the method is technically sound, the presentation is fair (while improvement on the readability can further enhance its quality), and the insights presented are of decent significance.

While there is some concern on the limited scope on insights, in that more insights than what’s already included in this paper will certainly boost the paper’s strength linearly, most think that the breadth of the insight coverage is significant enough. There is also one remaining concern on the LMC front, which questions the lazy regime hypothesis. While a proxy model would certainly exhibit some performance gap, the insights gleaned from it (and explanations derived) can still contribute positively to one’s understanding of the original model. The only reviewer that recommended against accepting seems to fixate on 1 case study only and shows low confidence on the other 2.

Therefore I choose to side with the majority of reviewers and recommend accepting.